# Conformer-dependent vacuum ultraviolet photodynamics and chiral asymmetries in pure enantiomers of gas phase proline

Rim Hadidi [1], Dušan K. Božanić, [1,3], Hassan Ganjitabar [2], Gustavo A. Garcia [1], Ivan Powis [2] & Laurent Nahon [1✉]

Proline is a unique amino-acid, with a secondary amine fixed within a pyrrolidine ring providing specific structural properties to proline-rich biopolymers. Gas-phase proline possesses four main H-bond stabilized conformers differing by the ring puckering and carboxylic acid orientation. The latter defines two classes of conformation, whose large ionization energy difference allows a unique conformer-class tagging via electron spectroscopy. Photoelectron circular dichroism (PECD) is an intense chiroptical effect sensitive to molecular structures, hence theorized to be highly conformation-dependent. Here, we present experimental evidence of an intense and striking conformer-specific PECD, measured in the vacuum ultraviolet (VUV) photoionization of proline, as well as a conformer-dependent cation fragmentation behavior. This finding, combined with theoretical modeling, allows a refinement of the conformational landscape and energetic ordering, that proves inaccessible to current molecular electronic structure calculations. Additionally, astrochemical implications regarding a possible link of PECD to the origin of life's homochirality are considered in terms of plausible temperature constraints.

[1] Synchrotron SOLEIL, l'Orme des Merisiers, Gif sur Yvette Cedex, France. [2] School of Chemistry, The University of Nottingham, University Park, Nottingham, UK. [3] Present address: Department of Radiation Chemistry and Physics, "VINČA" Institute of Nuclear Sciences—National Institute of the Republic of Serbia, University of Belgrade, Belgrade, Serbia. ✉email: laurent.nahon@synchrotron-soleil.fr

Proline (Pro) is the only proteic α-amino-acid containing a secondary amine (N–H) fixed within a pyrrolidine ring, which makes it conformationally less flexible than most other amino-acids. Owing to its specific structure, Pro plays an important role in determining the structures of proteins and peptides. Pro is indeed strongly involved in their formation and alteration, as its cyclic nature constrains the peptide backbone[1], providing Pro-rich proteins with very specific and important structural properties, for example in Intrinsically Disordered Proteins[2] and in collagen[3].

In the gas phase, on which we focus from now, the study of elementary building blocks of life such as amino-acids is at the basis of the so-called bottom-up approach of biomolecular complexity[4]. The gas phase offers a solvent-free and substrate-free environment, so that all intermolecular interactions, including those with a solvent or substrate, can be neglected; only intramolecular interactions are to be taken into account, such as non-covalent bonds responsible for conformations. In such a context, molecules can be studied in detail within an optimized interplay between experiment and theory. Besides, dilute matter may be probed by photons over a wide "transparent" spectral range, including the vacuum ultraviolet (VUV).

A motivation for the current work has been the continuation and extension of our previous studies on the VUV photoionization of alanine enantiomers[5,6], which adduced a potential new scenario for postulated astrophysical origins of life's homochirality[7]. Both alanine (Ala) and proline (Pro) belong to the first five amino-acids to have been recruited into the genetic code[8], and have also been detected in the Murchison meteorite in large quantities and with an excess of the L enantiomer[9]. Hence it will be of considerable interest to seek to establish whether similar properties apply for the chiral VUV photoionization of Pro, as was tentatively suggested[10]. In dilute environments such as the interstellar medium, Pro, like other amino-acids, is to be expected in its neutral form, unlike the zwitterionic forms found in condensed phases. Hence molecular structure of amino-acids in the gas phase requires its own direct study.

The molecular and electronic structures of gas phase neutral Pro have been the subject of a large number of experimental studies based upon IR (in a matrix)[11,12], microwave[13,14] and photoelectron[15–17] spectroscopies, as well as theoretical modeling[18–24]. In addition, the VUV[25,26] and soft X-ray[27,28] photoabsorption and photoionization of proline were studied in the gas phase by mass spectrometry, as well as its fragmentation by electron impact[29–31]. However, as far as we know, no photoelectron/photoion coincidence (PEPICO) studies have been done to measure state-selected fragmentation. Moreover, and most of all, no gas phase chiroptical study of any kind on this major amino-acid is available in the literature.

Most of the previous works considered the presence of four main low-energy conformers, divided into two groups (I and II), differing by their intramolecular hydrogen bond network. The strength of the H-bond interactions, especially in the neutral[21], leads to a surprisingly large difference in the ionization energy (IE) of the highest occupied molecular orbital (HOMO) of ~0.7 eV[15] between group I and group II conformers. By using Photoelectron Spectroscopy (PES) as a conformer tag, the rich and unusual conformer landscape of Pro allows us to template various photon-induced processes in a conformer-specific way, such as the VUV photodynamics (state-selected ion fragmentation) and most of all chiroptical electron asymmetries induced by the so-called photoelectron circular dichroism (PECD) effect.

PECD is an electric-dipole allowed, orbital-specific and photon energy-dependent effect, observed when pure enantiomers, randomly oriented in the gas phase, are ionized by circularly polarized light (CPL), this leading to a forward/backward

asymmetry (with respect to the photon axis) in the photoelectron angular distribution[32]. More precisely, the normalized photoelectron angular distribution for one-photon ionization takes the form $I_p(\theta) = 1 + b_1^{\{p\}} P_1(\cos\theta) + b_2^{\{p\}} P_2(\cos\theta)$, where $P_j$ are the Legendre polynomials, $\theta$ is the direction of the emitted electron and $p$ the polarization of the ionizing radiation ($p = 0, +1$ and $-1$ for linear, left circular and right circular polarizations, respectively). For CPL, $\theta$ is measured from the photon propagation axis. The so-called dichroic parameter $b_1$ is non-zero only for chiral systems photoionized with CPL and is antisymmetric under swapping of either light helicity or enantiomers. PECD is defined as $2b_1^{\{+1\}} P_1(\cos\theta)$, which is just the difference between the angular distributions obtained with left and right circular polarization radiations, corresponding to a maximum asymmetry of $2b_1^{\{+1\}}$ in the forward−backward direction. Therefore, $b_1^{\{+1\}}$ encapsulates the chiral contribution as well as the whole dynamics of the departing photoelectron scattering off an intrinsically asymmetric potential. The $b_2^{\{p\}}$ parameter expresses the symmetric part of the angular distribution, and for linear polarization $b_2^{\{0\}} \equiv \beta$, the familiar anisotropy parameter.

Since PECD is fully developed in the electric dipole approximation, it leads to very intense asymmetries, in the few %-few tens % range up to 37% as recently measured[33], which makes it very well adapted to low-density media such as the gas phase. Besides, PECD has been shown to be very sensitive to static molecular structures, such as conformers, isomers, clusters, chemical substitution and to dynamic molecular motions such as vibrations (for reviews see refs. [10,34]).

Because of the specific sensitivity of PECD to conformations[35,36] (for a review see also ref. [37]), there is a considerable interest in applying PECD to study biomolecules, especially amino-acids[5,6], which are known to possess a broad conformational landscape of crucial importance for the building up of larger biopolymers such as peptides and proteins. However, so far, all PECD studies of floppy systems possessing several conformers have been based upon a Boltzmann-averaged distribution of all conformers. In this context, the easily separable conformer-specific IE of Pro offers a unique opportunity to retrieve experimental conformer-specific PECD data to benchmark the theoretical modeling. Conversely, careful experiment/theory interplay may allow the use of PECD to refine the conformational landscape of Pro, especially in terms of energetics. Note that while conformer-specific CD in the ion yield has already been demonstrated via a resonant multi-photon ionization (REMPI) scheme[38], so far no corresponding conformer-specific REMPI-PECD has been reported although the needed high spectral resolution ns-REMPI PECD has recently been demonstrated[39].

Here, we present a comprehensive valence-shell photoionization study over a broad VUV range (8.7–17.5 eV) on Pro brought into the gas phase via two complementary vaporization methods. This study includes PES, state-selected fragmentation patterns and chiral asymmetries (PECD), as measured by double imaging PEPICO (i²PEPICO) and supported by dedicated theoretical calculations. By varying the experimental temperature, and consequently changing the Boltzmann conformer population we attempt to refine the conformational landscape of Pro and to benchmark scattering models on individual conformers. Finally, the astrophysical PECD-based scenario for the origin of life's homochirality is addressed, with this conformer-dependent analysis on Pro used to examine possible temperature constraints and implications relevant to interstellar medium conditions.

## Results and discussion

We used two complementary vaporization methods to bring Pro into the gas phase: Resistive Heating (RH) associated with an adiabatic expansion and aerosol ThermoDesorption (TD) at two

**Table 1 Summary of calculated energetics/ionization energies/geometry for the four main conformers of proline.**

| | Conformer type II | | Conformer type I | |
|---|---|---|---|---|
| | A | B | C | D |
| **Vertical ionization energy (eV)** | | | | |
| This work | 9.581 | 9.717 | 9.022 | 9.054 |
| Lu et al.[24] | 9.49 | | | 8.91 |
| Tian et al.[21] | 9.41 | 9.52 | 8.71 | 8.83 |
| Fathi et al.[16] | 8.81 | 8.941 | | |
| Dehareng et al.[19] | 9.36 | | 8.75 | |
| **Adiabatic ionization energy (eV)** | | | | |
| This work | 8.75 | 8.68 | 8.30 | 8.36 |
| Lu et al.[24] | 8.61 | | | 8.16 |
| **H-bond length in neutral (Å)** | | | | |
| This work | 1.828 | 1.864 | 2.404 | 2.237 |
| Czinki et al.[20] | 1.877 | 1.898 | 2.248 | 2.363 |
| Tian et al.[21] | 1.869 | 1.885 | 2.365 | 2.243 |
| Lesarri et al.[14] | 1.915 | | | |
| **NCC=O Dihedral angle in neutral [°]** | | | | |
| This work | −177.4 | −165.3 | −1.1 | −9.8 |
| Lesarri et al.[14] | −180 | −180 | | |
| **Relative energy (kJ mol$^{-1}$)** | | | | |
| This work(a) | 0.0 | 3.59 | 6.91 | 6.59 |
| Czinki et al.[20](b) | 0.0 | 1.98 | 7.66 | 8.56 |
| Mata et al.[13](c) | 0.0 | 3.21 | 8.91 | 9.35 |
| Fathi et al.[16](d) | 0.0 | 1.13 | 4.22 | 3.81 |

Results obtained in this work are from G3 composite method calculations except for vertical ionization energies, which are OVGF/cc-pVTZ//MP2/cc-pVTZ calculations. The relative energies are evaluated at 415 K, everything else assumes a temperature of 0 K.
(a)$\Delta G$ (G3) at 415 K (see Methods).
(b)$\Delta E$ (B3LYP level).
(c)$\Delta E$ MP2/6-311++G(d,p).
(d)$\Delta E$ MP2/6-311++G(d,p) at 403 K.

In Fig. 1, we show the geometric structure of the four proline conformers, with the two alternative H-bonding arrangements (Type I, Type II), and the two alternative up-down ring puckerings. Unfortunately, the labels $I_a$, $II_b$ …etc… have not been used consistently in the literature. We therefore choose labels A – D for the individual conformers to avoid this ambiguity, and in Table S1 (supplementary Information) we provide a table showing the correspondence with the various labeling choices used by other authors.

In Table 1, we gather various calculations of Pro conformer geometry and energetics from previous studies and compare these with our present set of G3 calculations (see Methods). The conformer energetics are shown schematically in Fig. 1. The calculations consistently show that the N··· H–O H-bonding interaction is the energetically most favorable (conformers Type II: A,B). Table 1 also shows that there is generally a very small energy difference between the calculated Type I conformers (C, D), with some corresponding controversy concerning the relative stability of C, D. This is to be expected since the predicted energy differences of ~1 kJ mol$^{-1}$ are at the limit of achievable computational accuracy.

**Photoelectron spectroscopy**. The first threshold electron spectroscopy and state-selected fragmentation measurements, using the TPEPICO technique, were performed for proline using the three vaporization conditions. These are presented in Fig. 2. In the top panel, we show the TPES spectrum of Pro recorded between 8 and 11.6 eV (up to 10.1 eV only for the RH condition), obtained in coincidence with the total mass (i.e., sum of the parent and fragment masses).

Two main bands are observed in our TPES, in good agreement with the PES of the literature[15–17]. The first broad band in the 8.1–10 eV energy range, corresponds to ionization from the highest occupied molecular orbital (HOMO)[15]. As seen in Fig. 3 (and the Mulliken population analysis provided as Table S2 in the Supplementary Information), for the Type I conformer pair this has the expected nitrogen lone pair character ($n_N$). However, the Type II conformers are now seen to have HOMOs that are more mixed in character. Although not shown here, a similar distinction can be made for the HOMO-1 orbital characters of these conformers, so that the second PES band, centered around 10.7 eV corresponds to ionization of an oxygen lone pair orbital ($n_O$) from the carbonyl group (C=O) of the Type I conformers, but more mixed and delocalized orbitals around the NH–C–COOH grouping in case of the Type II conformers.

With increasing temperature conditions (TD$_{415}$, TD$_{493}$, and then RH), three main effects are induced in the observed TPES (Fig. 2): (i) a clear shape change with the appearance of a broad shoulder around ~8.8 eV. This feature was already reported by Plekan et al.[15], and attributed to the increased population of conformers C and D possessing the lowest IE$_{vert}$; (ii) an apparent shift (from 9.5 to 9.65 eV) of the maximum of the HOMO peak for the RH case relative to the TD cases. This behavior could be due, in such a hot condition (see below), to a population increase of neutral conformer B with respect to the conformer A, which possesses a higher IP$_{vert}$ (by ~0.15 eV) than conformer A (see Table 1); (iii) a slight red shift of the spectrum probably due to hot bands in the neutral. This last feature is better appreciated on the lower panel of Fig. 2, showing the parent-filtered TPES (m/z 115) on which the apparent ionization thresholds under the three vaporization conditions are marked as the first point to rise above the baseline. These values are listed in Table 2 for each experimental condition. Note that the TD$_{415}$ threshold value of 8.30 ± 0.01 eV matches well the calculated adiabatic ionization energies of 8.30 eV and 8.36 eV for conformers C and D,

different temperatures, which leads to the three vaporization conditions of the neutral labeled as RH, TD$_{415}$, TD$_{493}$, respectively. After ionization of these neutrals by VUV synchrotron radiation (SR), the corresponding electrons and ions are detected in coincidence by imaging (time and position sensitive) detectors. By considering only threshold electrons (i.e., those with nearly zero kinetic energy), while scanning the photon energy, one may retrieve the total Threshold PhotoElectron Spectrum (TPES) by analysis of the electron image. By selecting only those electrons coincident with a chosen ion mass the so-called Threshold PhotoElectron/PhotoIon COincidence (TPEPICO) spectrum provides a state-selected fragmentation pattern. Considering instead all (i.e., fast and slow) electrons collected at a fixed-photon energy, our set-up provides, from the radial and angular distribution observed in a (mass-filtered) electron image, the angle-resolved photoelectron spectrum (PES). From pairs of images recorded with left- and right- circularly polarized light the chiral angular parameter, $b_1^{\{+1\}}$, can be extracted as a function of electron kinetic energy (for more details see the Methods section).

**Conformational landscape**. In the first microwave study of the proline rotational spectrum, only two conformers were detected, denoted II$_a$ and II$_b$[14]. A short distance between the nitrogen atom and the hydrogen of the carboxyl group supports the prediction of an N··· H–O hydrogen bond interaction, similar to conformation II found for all aliphatic α-amino-acids[19], with II$_a$ and II$_b$ differing by the pyrrolidine ring puckering. Then, a further MW experiment identified two other conformers I$_a$ and I$_b$[13], which are NH… O=C hydrogen bonded, this second pair again differing by the pyrrolidine ring puckering. Other works on proline confirm that energetics, H-bond strength (see Table 1) and population of both types of conformers supports the preference for the N··· H–O interaction[15,16,19,21] (Type II), as the most stable conformer type, due to geometric constraints imposed by the pyrrolidine ring.

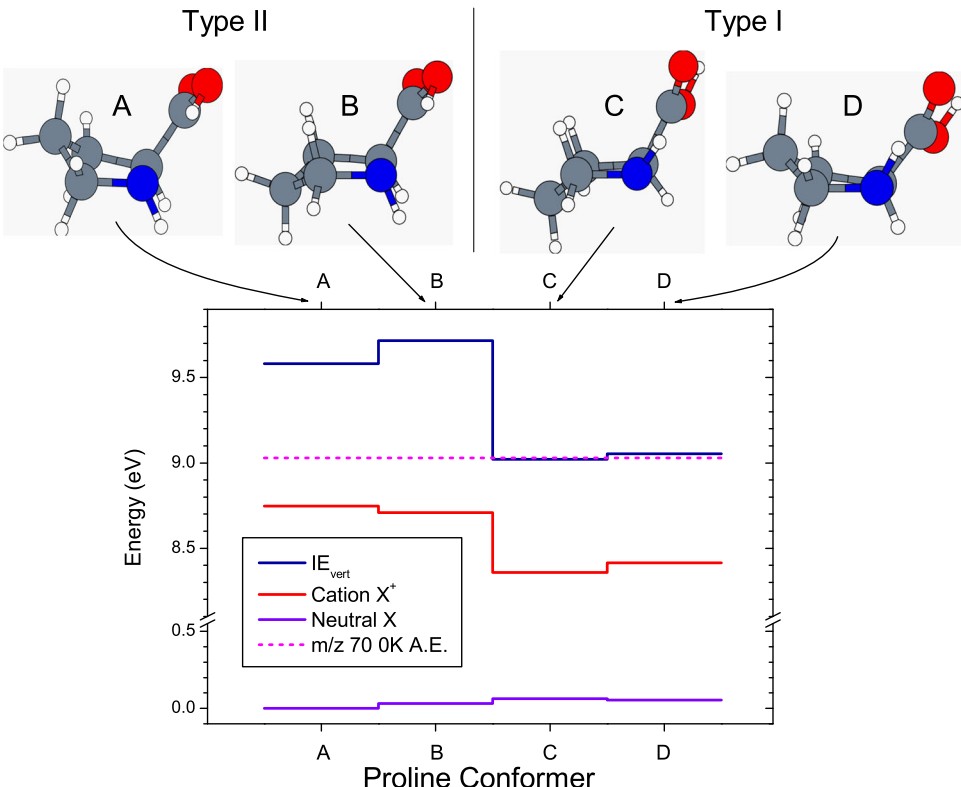

**Fig. 1 Calculated proline conformer structure and energetics.** The O atoms are colored in red, the N atoms in blue. The blue curve represents the electronic energies of the neutral and the red one the energies of the cation, with the difference being the estimated adiabatic (0 K) ionization energy (see Table 1). The vertical ionization energies (IE_vert) appear as a dark blue curve. Also shown (broken magenta line) is the experimentally estimated 0 K appearance energy of the principal fragment $m/z$ 70.

respectively (see Table 1). We thus concur with the conclusion of Plekan et al.[15] that in the low-energy side of the HOMO TPES band one selectively observes ionization of the thermodynamically less stable Type I conformer structures C and D. We also recorded the PES of Pro at different photon energies in $TD_{415}$ condition (Fig. S1 and S2). These show inner orbital ionizations with a peak centered around ~11.7 eV (shoulder) attributed to the HOMO-2 orbital, ($\pi_{CO}$) of the carbonyl group, while the peak located around ~12 eV is attributed to the HOMO-3 orbital, ($\sigma_{CC}$) orbital. Overall, our experimental results are in a good agreement with the calculated ionization energies of Pro, as reported in Table S3.

**Appearance energies and fragmentation behavior.** In Fig. 2 (lower two panels) are shown the parent-selected ($m/z$ 115) and fragment-selected ($m/z$ 70) TPES for the three conditions $TD_{415}$, $TD_{493}$, and RH. By correlating the fragmentation pattern with the electronic structure at different temperatures, a significant temperature effect is visible on these fragmentation diagrams. Quite generally, one may expect to observe increasing ion fragmentation (decreasing parent ion yield) with increasing internal vibrational excitation (temperature) in the neutral at the instant of its ionization[40]. Correspondingly, there is an expected apparent reduction in the fragment appearance energy with increasing thermal excitation in the neutral. Indeed, when looking at the trend of the $m/z$ 70 fragment curve, we note a clear red shift of the appearance energy of this fragment in the sequence $TD_{415}$, $TD_{493}$, RH from, respectively, 8.6, 8.4 to 8.3 eV. Noting also an increasingly prominent TPES shoulder at ~ 8.8 eV, which as mentioned above is expected to show such an increase with increasing temperature, one deduces that although the oven was heated only to 468 K in the RH

method, the RH source apparently produces much hotter neutrals than does the TD heated at 493 K.

To get a more precise knowledge of the actual temperatures of the neutrals, we modeled, for each vaporization condition, the breakdown diagram, i.e., the normalized fragment-to-parent abundance, as a function of photon energy (Fig. S3). This can be described using a statistical model assuming that all internal degrees of freedom are thermalized and that the energy stored in these modes can flow freely to be used, for instance, to excite a particular vibrational mode leading to fragmentation[41,42]. We assume that the fragmentation is faster than the typical residence time of the ion in the acceleration region (~µsec). The output of the model fit (Fig. S3), internal temperatures and the corresponding 0 K appearance energies (AE) that are deduced, are summarized in Table 2.

For the two TD conditions, the observed internal energy corresponds to temperatures, which are between 30 and 40 K colder than the TD tip, which is consistent with our past experience with alanine[6]. This decrease of temperature could be due to the slight cooling of the TD tip due to the sublimation process, or to some mild expansion of the plume. Contrastingly, the apparent temperature for the RH source is found to be 600 K, considerably above the actual temperature read by the thermocouple of the oven (468 K). We attribute this mismatch to the non-equilibrium conditions in the supersonic expansion, rendering the use of a single descriptive "temperature" rather questionable and/or poor location of the thermocouples on the oven body giving misleading temperature readings.

Finally, we note that the 0 K AE of $m/z$ 70 fragment is determined here for the first time to the best of our knowledge, with a good precision as confirmed by the very close values obtained from two fits providing a mean value of 9.02 eV.

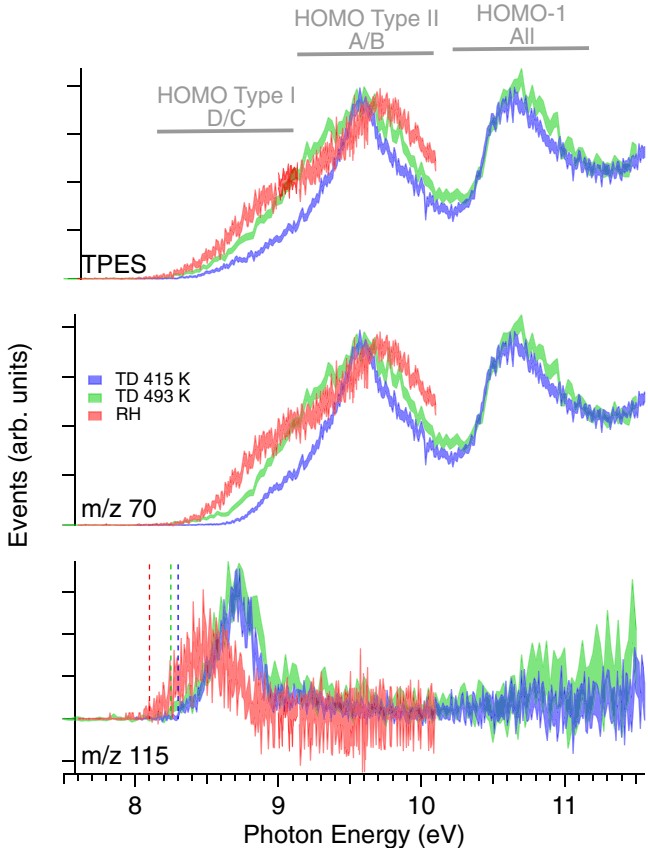

**Fig. 2 TPES and TPEPICO of Pro obtained for the 3 vaporization conditions.** The relative signals have been normalized according to the maximum of the TPES first band centered around 9.5 eV. The shaded area correspond to the (standard deviation) error bars for acquisition time ranging from 85 s/points (TD$_{493}$) to 160 s/points (RH). The dashed lines in the lower panel show the estimated apparent ionization energy obtained from the $m/z$ 115 TPEPICO curve as the first point to rise above the baseline. While the upper panel shows the full TPES, obtained by recording all threshold electrons—those having approximately zero kinetic energy such that all photon energy in excess of the ionization limit is deposited as internal excitation of the cation—the lower panels only show those threshold electrons that are found by TPEPICO coincidence detection to be accompanied by production of either parent or fragment ions. Clear temperature effects are observed on the shape and onsets (see text) of these mass-selected TPES.

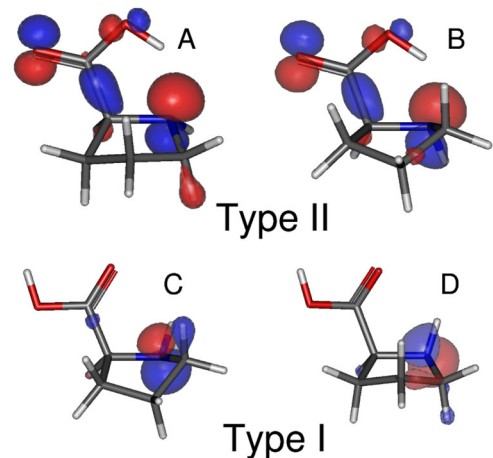

**Fig. 3 Hartree-Fock orbital density of the outermost (HOMO) orbital of the four proline conformers labelled A, B, C and D.** Type I conformer orbitals appear very localized at the N atom site (colored blue) but this is much less the case of conformer Type II orbitals showing more delocalization with some density also at the O atom sites (red) and, notably, around the C–Cα bond. These differences may partially account for the different fragmentation behavior between the two types of conformer.

vibrational modes in the cations can be expected, capable of driving the fragmentation process at the origin of the $m/z$ 70 fragment. This is especially the case for Type II conformers, since vertical excitation leaves the parent cation with excess energy above the fragment appearance energy as shown in Fig. 1.

Furthermore, the different orbital characteristics (Fig. 3) may be considered. The N lone pair HOMOs of Type I conformers can be expected to be essentially non-bonding, with minimal consequences for the C–Cα bonding upon ionization. In contrast, the more delocalized Type II HOMOs, that include density around the C–Cα bond are likely to lead to its weakening in the ion. Hence, it is very reasonable that close to their respective ionization thresholds the dissociative ionization behavior provides a signature of the two types of conformers: Type I conformers are partially stable while Type II ones are fully dissociative. The PEPICO scheme, by selecting parent ion production, can effectively isolate the specific conformer Type I (C/D), with a low-vibrational energy content, which is exploited in the forthcoming PECD astrophysical implications.

Ionization of the HOMO-1 orbital, commencing well above the fragmentation threshold, is fully dissociative leading solely to the $m/z$ 70 fragment. These features are also clearly visible on the mass-selected PES recorded at the 11.5 eV fixed-photon energy, shown in Fig. S2.

**PECD: experimental data and theoretical modeling.** Figure 4 shows photoelectron spectra (PES) and PECD curves as a function of the ionization energy (eV), measured for both L and D Proline enantiomers at different photon energies, in TD$_{415}$ conditions.

At 8.7 eV, the PECD of L and D enantiomers shows a clear and expected mirroring effect, with a magnitude of the order of 10 % for the low binding energy range that, as noted, corresponds to the pure conformer Type I (C/D) ionization. This value changes with the photon energy. At 9.5 eV photon energy, the PECD magnitude reaches 18 % in the low binding energy region, an asymmetry level unprecedented for a multi-conformer system. The higher binding energy part of the PES, corresponding predominantly to Type II (A/B) conformer ionization, exhibits a very different PECD level, close to zero. Therefore, the large IE

Returning to the TPES of Fig. 2, we notice that the HOMO orbital has a double-band structure whose shape changes with the temperature. This we attribute to the varying population of four main conformers populated in our experimental conditions. As already mentioned, and shown in Table 1, the first shoulder of the first band centered around ~8.7 eV, corresponds to the ionization of Type I conformer (C and D), while the second peak centered around 9.5–9.7 eV would correspond predominantly to the ionization of conformer Type II (A and B)[15].

It is clear that near their respective ionization thresholds, conformers Type I initially survive as a stable parent ion whereas Type II conformers undergo C–Cα bond breakup. This difference can be rationalized in more mechanistic terms by considering the much larger changes in geometry between the neutral and the cation, both in terms of H-bond length and NCC=O dihedral angles, predicted for Conformer Type II (A, B) than Type I (C, D) (see Table S4). In the vertical Franck-Condon approximation, the possibility for significant excitation of the corresponding

**Table 2 Extraction of temperature and *m/z* 70 fragment AE$_{0K}$ for the 3 vaporization conditions from a statistical modeling of the breakdown diagrams shown in Fig. S3.**

| Vaporization condition | Internal temperature deduced (K) | Fitted 0 K fragment (*m/z* 70) appearance energy (eV) | Expected Conf. I /Conf. II ratio | Ionization energy thresholds (eV) |
|---|---|---|---|---|
| TD$_{415}$ | 384 | 9.03 | 0.17 | 8.30(1) |
| TD$_{493}$ | 452 | -$^a$ | 0.24 | 8.25(2) |
| RH | 600 | 9.01 | 0.35 | 8.10(7) |

The 3rd column shows the Boltzmann expected Conf I/ Conf II population ratio based upon our calculated relative energies as shown in Table 1. The last column summarizes the apparent ionization energy threshold of the parent as deduced from Fig. 2.
$^a$For TD$_{493}$ the AE$_{0K}$ was held at 9.03 eV and only the slope was modeled.

difference between the two types of Pro conformers offers the opportunity to directly observe, for the first time experimentally to the best of our knowledge, a clear conformer specificity of PECD. At a photon energy of 11.5 eV (now for D-Pro) one notices a sign change of PECD for the low binding energy side (conformer type I region) as compared to the lower photon energy cases, as well as a different PECD level for the HOMO and HOMO-1 orbital. Such orbital and photon energy dependences are quite typical of PECD.

The PES and PECD of L/D-proline recorded by velocity map imaging at 10.2 eV photon energy in the three vaporization conditions (TD$_{415}$, TD$_{493}$, and RH) are shown in Fig. 5. The electron images used for these determinations have been mass-filtered on the parent (*m/z* 115) only, the fragment (*m/z* 70) only, and total mass (parent + fragment) ion coincidences, providing three alternative PECD curves for each sample condition. We note that the PES curve obtained by the RH method filtered on the parent has a maximum shifted by about 0.2–0.3 eV towards lower energies by comparison with the TD$_{415}$, a thermal effect already observed on the TPEPICO spectra of Fig. 2.

For the L-Pro TD$_{415}$ PECD (panel (a)), the Type I/Type II conformer-specificity is very clear already from the total mass curve, with an intense PECD of ~−10 % at ionization energies around 8.8 eV (Type I), switching sign to ~ +5% for ionization energies around 9.6 eV (Type II). This is even more pronounced when parent mass-filtered data are taken into account, with the selection of low-vibrational energy content only (IE~8.2–8.7 eV as seen from Fig. 2 bottom panel) for Type I conformers reaching about −12 % asymmetry. Besides, the PECD of Pro appears to be very sensitive to the temperature, in particular for the parent, whose PECD curve exhibits a lower magnitude at higher temperature: 12% for TD$_{415}$ and 6 % with the TD$_{493}$, and an average value of around 2–3% for the RH method. We attribute this change in magnitude to a relative change in the population of conformers C/D present in our experimental conditions, as we will further rationalize below.

Figure 6(a) and (b) shows the results of continuum multiple scattering (CMS-Xα) calculations on the HOMO photoionization observables made for the four proline conformers A, B, C, and D across a range of electron kinetic energy (KE). It is clear that the conformer sensitivity displayed by the $b_1$ parameter is dramatic, but less so for the cross-sections, $\sigma$. For low kinetic energy electrons, which are known to be the most sensitive to molecular structures[37], there are differences in the sign of $b_1$ predicted within both the Type II A/B and the Type I C/D conformer pairings, where the principal structural differences between those conformers of a given type are just the ring puckering. Equally, there are significant differences between the B/C pair and between the A/D pair. The latter, particularly, shows an enantiomer-like approximate mirroring of the A and D $b_1$ curves. Now, the principal structural change between A and D is a near 180º difference in the NCC=O dihedral angle (see Table S4), but retaining the same ring puckering.

It is interesting to compare with similar calculations made for the HOMO-1 ionization, as shown in Fig. 6(c), (d). The conformer-sensitivity of the $b_1^{\{+1\}}$ parameters is now much less pronounced despite the HOMO-1 orbitals retaining quite similar characteristics to the HOMOs—Type I localized lone pairs (however on O instead of N for the HOMO orbital), Type II delocalized around the N-C-COOH grouping—and clearly sharing the same nuclear geometries.

Focussing now on the HOMO ionization, each conformer $b_1$ parameter in Fig. 6(b) individually shows variations in sign and magnitude that are commensurate with the experimental variability noted above. However, a more detailed comparison of these theoretical predictions and experimental results is essential. We do so by extracting characteristic PECD values for the Type I and Type II conformer regions across a range of experimental photon energies using a simple, approximate deconvolution procedure. First two Gaussian functions are fitted to the PES as illustrated in Fig. 5(a). Weighted averages of the corresponding PECD (=2$b_1$) are then formed across the full width at half maximum (FWHM) of each Gaussian function as an attempt to isolate and reduce the data for Type I and Type II conformer regions.

Type I (C + D) mean $b_1^{\{+1\}}$ values that have been extracted from the low binding energy region using a gaussian weighting function FWHM of 0.6 eV and centered at ~8.75 eV ionization energy, recorded using photon energies ranging from 8.7 to 17.5 eV, are shown in Fig. 7. Also included in this figure are the calculated PECD curves for the Type I conformers, taken from Fig. 6(b).

Overall, the $b_1$ experimental data exhibits a very dynamical behavior with large variation and sign changes with the photon (equivalent electron kinetic) energy, reflecting the quantum nature of the scattering process from which PECD originates. On examination, it is readily seen that the experimental data are essentially bounded between the theoretical curves. Moreover, while the experimental data fall closer to the theoretical curve for conformer C, there is a clear temperature dependence with an increasing displacement of the TD$_{493}$ and RH data sets towards the conformer D prediction. Qualitatively, this appears to argue strongly for the C conformer being the more stable of the two, with an increasing population of the D conformer with increasing temperatures.

To pursue this more quantitatively, we have taken an expression,

$$b_1(\text{expt}) = x \times b_1(C) + (1 - x) \times b_1(D),$$

that blends the C and D conformer curves, where $x$ and $(1 − x)$ indicate the relative proportion of each conformer in the mix. By applying a least squares fit for $x$ to each data set, estimates for the C conformer relative populations of 78%, 74%, and 58% for, respectively, the TD$_{415}$, TD$_{493}$, and RH data sets are extracted (Note that KE = 1.5 eV point for the TD$_{415}$ set has been excluded

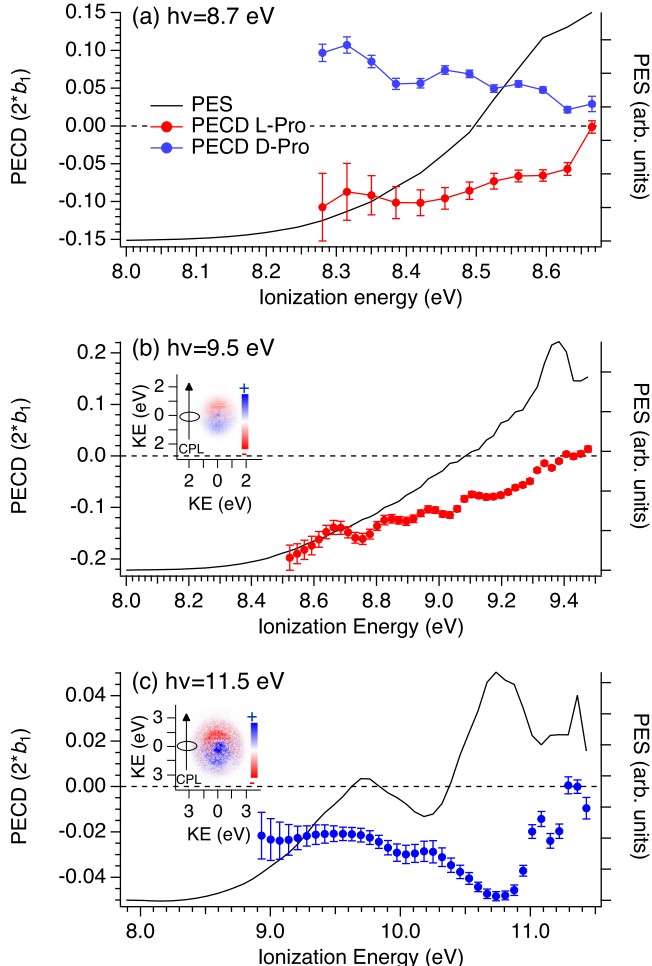

**Fig. 4 PES (solid curves) and PECD (symbols with error bars) obtained with D- (blue) and L- (red) proline enantiomers, produced in TD$_{415}$ conditions.** The data are extracted from a given pair of electron images recorded with left- and right- circularly polarized light (CPL) using photon energies of **a** 8.7 eV; **b** 9.5 eV; **c** 11. 5 eV. PECD points corresponding to a relative PES intensity below 10% of the maximum signal are not shown, due to their increasing error bars when normalized by the PES intensity. Error bars correspond to statistical (standard deviation) errors. The images include electrons coincident with formation of all masses associated with the ionization of Pro ($m/z$ 115 + $m/z$ 70). PES profiles can be understood as relative cross-sections for ion formation and at a given ionization energy will correspond to formation of a specific excited state of the ion; however, as photon energy is increased, the kinetic energy for electrons produced for a given ionization energy (state) will be likewise increased. A clear dependence of $b_1^{\{+1\}}$ with both the ion state and the associated electron kinetic energy is thus observed. Note the satisfactory and expected L/D mirroring in panel **a**. Inserts show raw difference photoelectron velocity map images obtained by subtracting left- and right-CPL measurements. The images show a forward/backward asymmetry with respect to the CPL propagation axis.

from the fitting as an outlier). Going further, the C:D population ratios may be equated to a Boltzmann factor, and using the inferred sample temperatures for each data set (Table 2) we deduce an energy difference, $\Delta E (= E_D - E_C)$ of 4.02±0.07 kJ mol$^{-1}$, 3.91±0.11 kJ mol$^{-1}$, and 1.55±0.29 kJ mol$^{-1}$ for, again, the TD$_{415}$, TD$_{493}$, and RH data sets.

The $\Delta E$ value deduced from the RH data set appears unexpectedly lower than the TD derived values. The quoted error estimates reflect only the statistical uncertainty from the

fitting procedure, but do not include more systematic errors. The small variations around 1 in the relative conformer cross-sections (see Fig. 6(a)) imply variations in the relative sensitivity to these conformers across the energy range but this has not been incorporated into the fit. The shape and position of the gaussian sampling function may also inadvertently induce some bias against one or the other conformer—in particular the ~0.2 eV shift noted for the parent mass-filtered RH TPES and PES (Figs. 2 and 5) probably indicates that a more sophisticated algorithm is required for consistent sampling of both RH and TD data sets.

Overall, however, Fig. 7 demonstrates that a convincing agreement between theory and experiment is achieved for the Type I conformers, with the C:D population differing in an understandable manner with the various sample inlet conditions. Qualitatively, it clearly shows that conformer C is the more stable, being more strongly populated under cooler sample conditions. Various calculations of the C and D conformer energies, reported in Table 1, are inconsistent in predicting the energy ordering of these two conformers—understandably since the calculated energy differences of ≲ 1 kJ mol$^{-1}$ are below the expected limit of achievable computational accuracy. While several caveats apply to our quantitative experimental estimates of the energy differences, they suggest the D conformer lies somewhat higher in energy, ~1–4 kJ mol$^{-1}$ above C. We are thus able to refine the conformer landscape and conclude unambiguously that conformer C is more stable than conformer D by a few kJ mol$^{-1}$.

A similar attempt was made to examine the Type II A/B conformer pair using a gaussian sampling function centered at 9.65 eV ionization energy to attempt to exclude Type I conformer ionizations. Figure 8 then provides an analogous plot to Fig. 7 for the A/B conformers, but with an evidently less satisfactory outcome. As already identified, the PECD asymmetries recorded in the Type II conformer region are of a lesser magnitude. While below 3 eV electron energy this could be a result of partial cancellation of the larger, but opposing PECD calculated for conformers A and B, in the mid-range the experimental values remain small but positive, falling outside the range of negative values bounded by the theoretical predictions.

One possible explanation could be that the gaussian sampling method is unable to exclude contributions made by Type I conformers in this higher binding energy range. For example, we note that the conformer D PECD, in particular, is strongly positive across this central kinetic energy region, which could act to counter any negative A/B. However, a preponderant contribution by D conformer in this region is unlikely in consideration of predicted conformer populations. Moreover, given D conformer has a non-bonding $n_N$ HOMO, its ionization is unlikely to provide a Franck-Condon envelope that extends much beyond its ionization threshold. Alternatively, we may consider possible limitations of the CMS-Xα method, particularly in relation to the A/B conformer calculations. Experience has shown that such PECD calculations perform well for localized initial orbitals—such as 1$s$ core orbitals[43,44] and lone pair valence orbitals[45,46] and now, evidently, including the C/D conformer HOMO orbitals of Pro. In contrast, less well localized valence orbitals, such as the HOMO orbitals of the A and B conformers (see Fig. 3), can sometimes pose a greater challenge. This follows from the manner in which the Xα potential is partitioned into spherical atomic regions. The spatial extent of core and lone pair orbitals is largely confined to one such spherical region, whereas an adequate description of more delocalized bonding orbitals would require additional modeling of the internuclear electron density. Overlapping adjacent spherical regions can partially assist in reliably modeling such non-local electron density, but specifically in the Type II conformers these challenges are compounded by the particularly strong H-bonds influencing electron density

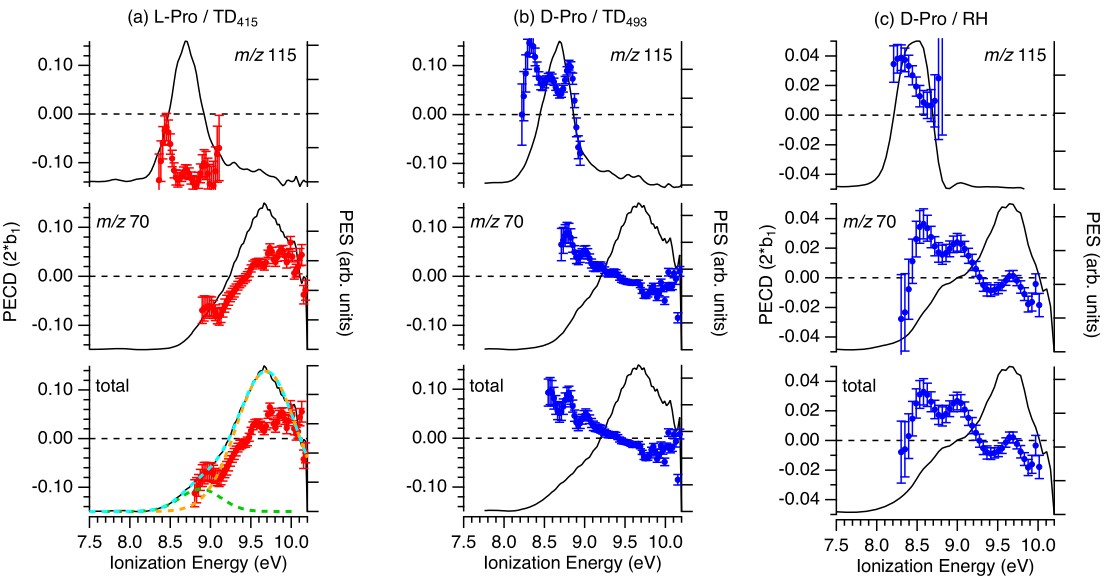

**Fig. 5 PECD (red markers L-Pro, blue D-Pro) and PES (black curve) of proline recorded at 10.2 eV for the 3 vaporization conditions.** The data are filtered on the parent ($m/z$ 115) and the fragment ($m/z$ 70) as well as the total mass. **a** $TD_{415}$ (L-Pro). **b** $TD_{493}$ (D-Pro). **c** RH (D-Pro). PECD points corresponding to a relative PES intensity below 10% (15% for the $m/z$ 115 $TD_{415}$ plot) of the maximum signal are not shown, due to their increasing statistical (standard deviation) error bars when normalized by the PES intensity. The total PES in panel **a** shows the deconvolution procedure too, with two Gaussian functions used to fit the PES (dashed cyan line), one for Type I centered at 8.75 eV (dashed green curve) and Type II centered at 9.65 eV (dashed orange curve) conformers. Note that data in panel **c** are displayed with a different vertical axis scale.

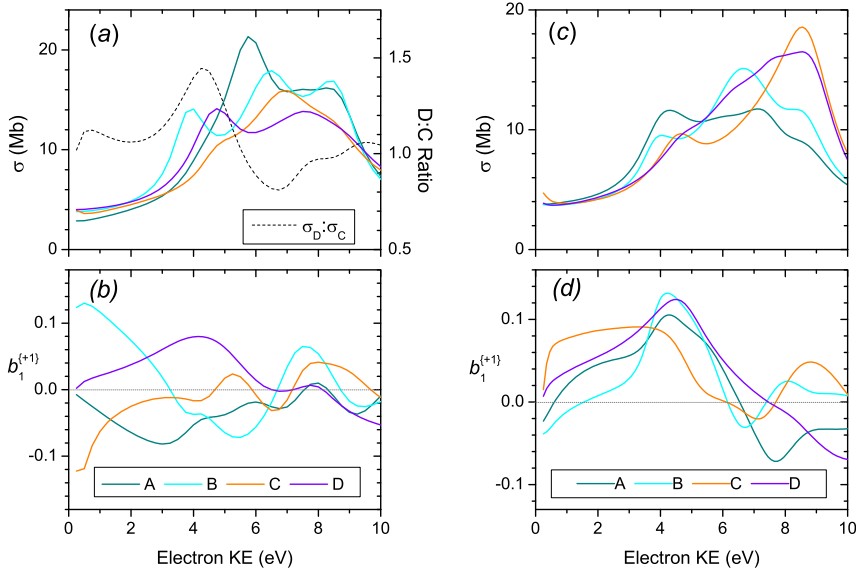

**Fig. 6 CMS-Xα calculations on L-proline for photoionization of conformers A–D. a** and **b** HOMO orbital; **c** and **d** HOMO-1 orbital. The top panels (**a** and **c**) show partial cross-sections, $\sigma$; the bottom panels (**b** and **d**) display chiral asymmetry parameters $b_1^{\{+1\}}$. Panel **a** for the HOMO photoionization also shows the D:C conformer cross-section ratio as a dashed line.

between N and H nuclear centers at intermediate ~1.8 Å separations. Permitting sphere overlap at this range would require over-extended atomic spheres, and cannot be readily accommodated. Such considerations do not appear to apply to the C/D conformers due to the much weaker N–H..O H-bonding.

Since there is no clear evidence of temperature-dependent behavior in the experimental data, this suggests that the A and B conformers may be nearly isoenergetic, with ~50:50 population. However, as we are unable to judge whether the poor theory-experiment agreement evident in the 3.5 eV–5.5 eV energy range of Fig. 8 is primarily experimental or theoretical in origin—or indeed is attributable to both causes—there are no more definite

conclusions to be drawn concerning the Type II(A/B) conformer energetics.

**Astrochemical implications.** The origin of life's homochirality, the fact that almost exclusively L-amino-acids in proteins, and D-sugars in nucleic acids, are found in the biosphere, is still a puzzling and open question, a cornerstone in modern science[7]. More than 170 years after Pasteur's first intuition, who saw in this biomolecular asymmetry a signature of life, as opposed to symmetric inert matter, the origin of homochirality and of life itself appears strongly entangled. Most scenarios, so-called abiotic,

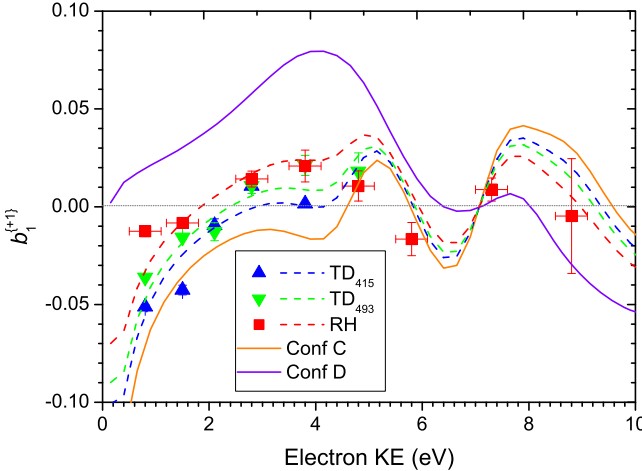

**Fig. 7 Type I conformer (C+D) $b_1^{\{+1\}}$ values for the HOMO orbital photoionization as a function of electron KE.** The mean experimental $b_1^{\{+1\}}$ data points, given by symbols, extracted from total (parent + fragment) mass-filtered TD$_{415}$, TD$_{495}$, RH data sets using a gaussian sampling function centered at 8.75 eV binding energy, are shown for L-Pro with any D-Pro data negated prior to plotting. Horizontal error bars are attached to the RH data only, representing the FWHM of the gaussian sampling function that isolates the Type I region; the same FWHM was also used for the TD data sets but for clarity has not been explicitly shown. Vertical error bars correspond to statistical (standard deviation) errors. Also included are solid curves showing the CMS-X$\alpha$ calculations for conformers C and D. Best fits to the experimental data sets obtained by combining these predictions (see text) appear as broken lines.

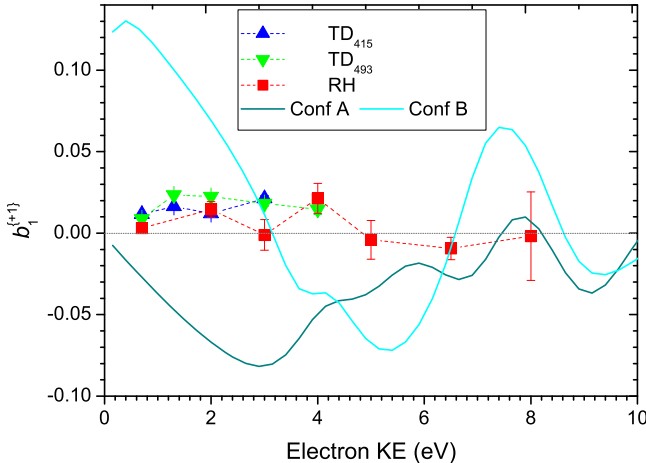

**Fig. 8 Conformer Type II (A+B) mean experimental $b_1^{\{+1\}}$ values for the HOMO orbital photoionization as a function of electron KE.** The data are extracted from total (parent + fragment) mass-filtered TD$_{415}$, TD$_{495}$, RH data sets using a gaussian sampling function centered at 9.65 eV binding energy. The mean experimental $b_1^{\{+1\}}$ data points are plotted at their central sampling energy for L-Pro, with any D-Pro data negated prior to plotting. Error bars correspond to statistical (standard deviation) errors. Broken lines joining data points are to guide the eye only. CMS-X$\alpha$ calculations for conformers A and B are shown as solid curves.

imply that homochirality, or at least a significant enantiomeric excess (ee), existed prior to the emergence of life and was even a necessary condition for its development, since a racemic life appears unlikely, for structural reasons in particular[47]. A very broad range of abiotic scenarios for the origin of biomolecular asymmetry have been proposed based either on random processes

or on deterministic ones involving chemical and/or physical forces (for a review see ref. [48]). The deterministic branch can be subdivided into scenarios based upon parity-violation in the weak interaction possibly leading to tiny energy differences between enantiomers[49,50], and those based upon the interaction with chiral fields such as magnetic fields[51] or the ones associated with CPL[52,53] on which we will focus from now.

Among the various media from which life might have originated, the circumstellar/interstellar medium (CSM/ISM) appear as very promising with the discovery of many proteic and non-proteic amino-acids, all with several % L-ee, in various meteorites including the well-known Murchison one[9,54–56]. These findings suggest an extra-terrestrial origin of life, more precisely that building blocks of life, such as amino-acids, would have been formed in the ISM and would have then been delivered to early Earth via meteorites and comets. During their journey towards Earth, they could have been submitted to a chiral bias, such as CPL, so that the organic material delivered on Earth would be enantio-enriched, before reaching homochirality (100 % ee) *via* autocatalytical reactions on Earth[57]. Such an astrophysical scenario is supported by the discovery of partially polarized CPL in large portion of space in heavy star formation regions such as the Orion Nebulae[58,59]. This major finding motivated several SR studies on UV/VUV asymmetric photochemistry in condensed matter[60–62] leading to a few % ee on amino-acids produced by asymmetric photochirogenesis from interstellar-analog ices[63,64] and by CD-driven asymmetric photolysis from racemic thin films[65,66].

In a previous study on alanine[5,6], we proposed PECD as an alternative asymmetric photophysical route in the gas phase for the production of ee in amino-acids in a given direction of space[10]. Indeed, gas phase amino-acids produced by evaporation from hot cores[67] or by desorption induced by photons or energetic particles from icy grains[68], a priori as a racemic mixture, might be submitted in the ISM to CPL irradiation in the VUV range, and especially at the strongly dominant Lyman-$\alpha$ radiation (10.2 eV)[64,69,70]. Because of the PECD effect, such an interaction would generate, for each enantiomer, asymmetric photoelectron angular distributions of opposite forward/backward asymmetry with respect to the photon axis. Then, because of momentum conservation, the corresponding ions would be recoiling in an opposite direction exhibiting a reverse angular bias for each enantiomer, as compared to the photoelectrons. In other words, PECD would produce two enantio-enriched hemispherical clouds of parent ions, recoiling one from the other in a given line of sight corresponding to the photon axis, with opposite ee. One of these two parent ion clouds, enriched with enantiomers of a given handedness, would have then been captured and embedded into a shower of meteorites and comets, which would have seeded early Earth with this enantio-enriched organic matter, triggering therefore life's homochirality.

In the case of Ala photoionized at Lyman-$\alpha$ radiation (10.2 eV), the filtered PECD ($2b_1$) on the parent mass ($m/z$ 89) reaches a value of 4% with a negative value for the L-enantiomer[5]. As the calculated conformer-resolved PECD at 10.2 eV for the 3 main conformers of Ala have the same exact value[6], it was not possible to give any constraints in terms of temperature since the PECD at this energy appears totally insensitive to Boltzmann averaging whatever the temperature.

The case of Proline offers a possibility to refine this PECD-based scenario as a possible effect involved in the origin of homochirality. Indeed Pro, as well as Ala, are both relevant for the origin of life having been one of the first amino-acid to be recruited by life in the genetic code, Pro being n°5 while Ala is n°2[8]. In a preliminary study, we showed that parent-filtered Pro ($m/z$ 115), produced by TD (TD$_{415}$) led to a PECD ($2b_1$) of 12 %, a threefold increase as

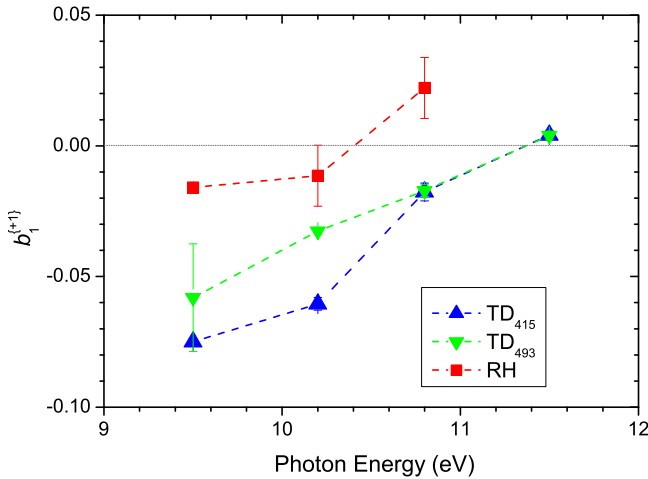

**Fig. 9 Parent mass-filtered $b_1^{\{+1\}}$ for L-pro HOMO ionization.** These are plotted as mean values, formed by PES intensity weighted averaging across the PECD profiles extracted from $m/z$ 115 filtered electron images. Error bars correspond to statistical (standard deviation) errors (for some points within the symbol size). Broken lines are to guide the eye only.

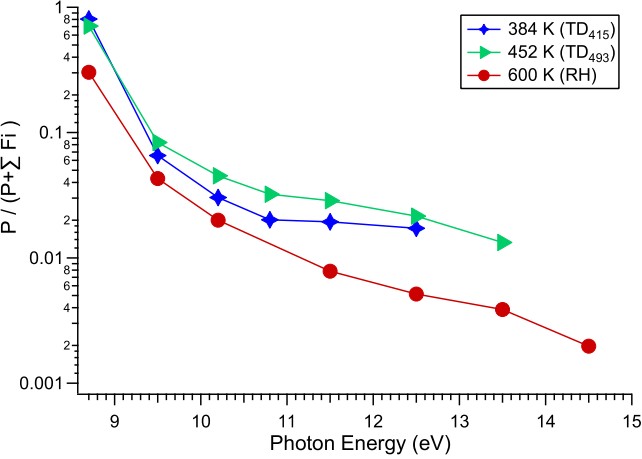

**Fig. 10 Survival rate ($P/(P + F)$) of the Pro parent (P) ion ($m/z$ 115) vs. fragment (F) ($m/z$ 70 + $m/z$ 43).** The data are shown as a function of the photon energy and temperature: 384 K (TD$_{415}$), 452 K (TD$_{493}$), and 600 K (RH). The data have been obtained from the mass spectra recorded during the PECD PEPICO acquisition. The statistical error bars (standard deviations) are within the symbol size.

compared to Ala, and most of all of the same sign (negative for L-enantiomer) as Ala[10] and most likely as Ser[71]. This sign matching is of course a crucial point for the validity of the scenario, which should bring the same ee sign for all amino-acids.

The Pro full PECD study at the core of the present paper allows further understanding of the astrochemical implications in terms of ee and abundance as a function of temperature. This is an important issue because a wide range of temperature can be found in the ISM ranging from a few K to several 100's K in hot cores[64].

Let us only consider here the data regarding conformer Type I (C and D) whose photoionization of the HOMO orbital is not dissociative, leading to intact parent ions, the only species to be considered for astrochemical implications. In Fig. 9 we show PECD data for several photon energies, extracted from parent mass-filtered profiles, such as shown for $h\nu = 10.2$ eV in Fig. 5. These data points display the same temperature dependence and follow the same trend with increasing energy as discussed for Fig. 7. However, the magnitude of the asymmetry at the lower energies studied is somehow increased compared to Fig. 7. This is attributable to the narrow electron energy selection implied by non-dissociative ionization subset included here, as opposed to the non-selective 0.6 eV FWHM gaussian used to obtain the Fig. 7 data. Such a KE effect, consequent on the skewing of the sampling function between these alternative selection strategies, is not surprising given the predicted rapid variation in conformer C's $b_1$ parameter near threshold (Fig. 6(b)). Experimentally also, this is explicitly visible in Fig. 4(b) ($h\nu = 9.5$ eV) and Fig. 5 ($h\nu = 10.2$ eV) where the most intense asymmetries correspond to the lowest binding energy range.

It is clear that overall the magnitude of PECD at 10.2 eV (corresponding to 1.5 eV KE for conformer Type I) decreases with increasing temperature, while keeping the same sign for a given enantiomer (always negative for L-Pro). Since both conformers at 10.2 eV (1.5 eV KE) appear to have a very similar computed magnitude with opposite sign, this leads to a decreasing PECD magnitude at very high temperature, down to ~2 % at 600 K (RH case). At the opposite, the extrapolation towards very low temperatures should lead to intense asymmetries, at least as high as the 12 % measured at 384 K (TD$_{415}$), only governed by the PECD of conformer C.

Of course, temperature might also have an impact on the abundance of parent cations, because of the temperature

dependence of ion fragmentation processes. In this respect Pro provides a very peculiar example. Indeed, as it can be seen in Fig. 10 showing the parent survival rate ($P/(P + F)$) as a function of the photon energy and temperature, the temperature effect is twofold: at low photon energy (8.7 eV) where mainly only Type I conformers can be ionized, the higher the temperature, the lower is ($P/(P + F)$), an expected behavior because of the increasing statistical dissociation in the hot ground state of the cation as observed for instance in Trp[72,73].

Above the IE of Type II conformers, the $P/F$ ratio is in addition governed by a Pro-specific competing conformer population effect, which at the opposite, tends to increase ($P/(P + F)$) with increasing temperature, by populating Type I (C/D) conformer in the neutral, which, as seen above, is partially non-dissociative. The combination of both processes leads for instance at 10.2 eV to an unexpected temperature ordering in which the highest survival rate (~5%) is found for the intermediate 452 K case (TD$_{493}$). At 10.2 eV, in the high-temperature limit, we found a survival rate of ~ 2% at 600 K. Extrapolation to very low temperatures, by depopulating the stable conformer Type I (C/D) should lead, non-intuitively, to a decreasing survival of the parent cation.

Overall, as it was the case of Ala, Pro finally does not show any temperature constraints for the considered astrophysical scenario, with a constant sign of PECD, and therefore of the ee sign of Pro parent cation recoiling in a given line of sight along the direction of the propagation of the CPL, whatever the temperature. At high temperature (say 600 K) the magnitude of the ee is reduced (to a few %), while it should reach high values at very low temperature (at least 12%), with however a decreasing abundance of parent ions. Note that in realistic ISM conditions, as for all the other CPL-based scenarios, these ee figures should be multiplied by the actual absolute circular polarization rate of the VUV radiation encountered in heavy star forming regions, typically lying in the few % to 20% range[58], leading to an effective ee in the 0.1% to 2% range.

## Conclusions

By using two complementary vaporization methods, TD and RH, coupled to a i2PEPICO detection scheme, we have been able to carry out a complete VUV photodynamics study on gas phase

pure enantiomers of the amino-acid proline, mainly focused on processes mediated by ionization from the HOMO orbital. Proline possesses four main conformers, populated in our experimental conditions, which can be divided into two groups I (C/D) and II (A/B) differing by the carboxylic group orientation versus the pyrrolidine ring, i.e., stabilized by different intramolecular H-bonds. Very interestingly, these two groups are associated with very different (by ~0.8 eV) IE offering a unique way to probe specific conformer dynamics when coupled to electron spectroscopy.

The TPES/TPEPICO analysis yielded an observed adiabatic IE of 8.30 ± 0.01 eV, (at 384 K) and a 0 K AE of the fragment $m/z$ 70, corresponding to a C–Cα bond cleavage, of 9.02 eV. Most of all, the TPEPICO spectra unraveled an unusual conformer-dependent fragmentation behavior of the state-selected cation, Type I conformer being partially stable, leading to a parent ($m/z$ 115) cation, while Type II is fully dissociative upon ionization because of a very large geometry change between the neutral and the cation. Such a specific fragmentation pattern is also clearly visible on the fixed-photon energy mass-filtered PES shown in Figure S2. Moreover, the fitting of TPEPICO provided us with the internal temperatures corresponding to the three vaporization conditions we used ($TD_{415}$, $TD_{493}$, and RH).

The PECD data appear as very structured, reaching very high values of up to 18% at a photon energy of 9.5 eV, which is quite unexpected for a multi-conformer molecule. Using the IE tagging to decipher conformers, it has been possible to observe directly a very strongly conformer-dependent PECD dynamics. Dedicated CMS-Xα calculations of the $b_1$ dichroic parameter indicate a striking conformer-dependence for the HOMO ionization in particular. These calculations for Type I(C/D) HOMO PECD clearly bound the corresponding experimental data, revealing also a clear C/D conformer temperature-dependent population changes across our experimental temperature range. Most strikingly, the C conformer is unambiguously identified as the more stable of the two, something which is not established by existing electronic structure calculations (including ours), due to the small energy differences that must be at the expected limit of computational accuracies. By fitting the PECD results for the three sample vaporization methods as a weighted average of the two theoretical predictions, quantitative C:D population ratios were extracted. By further taking the internal temperatures inferred for these inlet conditions, an energetic difference favouring the C conformer by a few kJ mol$^{-1}$ was estimated. This demonstrates that PECD, *via* its strong sensitivity to subtle details of the molecular potential, may allow refining the conformational landscape of floppy systems.

An analogous comparison of theory and experiment for the A/B conformer pair proved less successful. While the reasons for this remain to be fully established, it may be pertinent to note that while the C and D conformers have HOMOs possessing N lone pair character, the A and B conformer HOMOs have more delocalized character between and around the NH and COOH functional groups. Experience suggests that this non-local character may pose a greater challenge for the CMS-Xα theory. Experimentally also, the A/B PECD is weak, varying only slowly across the full energy range studied, and without any obvious temperature-dependent variation.

As an extension of this first chiroptical study of proline, and besides possible applications of our conformer-tagging method to the few other amino-acids possessing quite large Type I/Type II conformer IE separation (~0.3–0.6 eV)[23], future experimental efforts for a direct study of conformer-specific PECD on floppy systems could take two avenues: (i) using methods such as electrostatic deflection[74] to select a given conformer to be studied by one-photon PECD; (ii) using high spectral resolution REMPI-

PECD, as recently demonstrated[39], to tag a given conformer via the first exciting photon. Conversely, while having been predicted by theory as soon as year 2000[75] this unprecedented, to the best of our knowledge, experimental evidence of conformer-dependent PECD is important, considering the quickly growing field of table-top laser-based REMPI-PECD for in situ ee measurement in analytical chemistry[76,77] and probably soon analytical biochemistry for which conformational flexibility will be a challenging issue to deal with.

Besides, PECD has been suggested to possibly be involved in the origin of life's homochirality as an asymmetric photophysical process applied on gas phase amino-acids in the ISM, giving rise to enantiomeric excesses in a given line of sight of recoiling parent cations[10]. Such a scenario proposed for Ala as well as for Pro is strengthened here. Indeed, the present temperature-dependent PECD study, carried out at the astrophysically relevant Lyman-α radiation energy, does not set any constraints in terms temperatures, i.e., in terms of ISM/CSM types of environment for the validity of this scenario leading in all situations to the same sign of the ee for Ala and Pro, whatever the temperature. Such a thorough study should be extended in the future to other astrobiologically relevant amino-acids, as well as to the even more challenging case of nucleic acid sugars (ribose) for which, considering a given CPL helicity, the Lyman-α radiation PECD should exhibit a reverse asymmetry as compared to amino-acids, to account for the occurrence of only D-sugars in the biosphere.

## Methods
Except for a few recent improvements, the experimental set-up based upon the SAPHIRS versatile molecular beam chamber, a permanent endstation of the DESIRS beamline at Synchrotron SOLEIL (St Aubin, France), is quite similar to the one that has been used for Ala[5,6]. We used two complementary methods to bring the thermolabile Pro molecule into the gas phase: (i) resistive heating (RH), which might lead to severe decomposition/polymerization and (ii) aerosol thermodesorption (TD). The interplay of both methods allows us to benefit from their combined individual advantages and, in the present context, leads to different internal temperatures of gas phase Pro. In both cases, L- and D-Pro were purchased from Sigma-Aldrich (>99 % purity).

**Resistive heating vaporization method (RH).** A few grams of Pro powder were nested into several layers of fiberglass wool and placed in the metallic reservoir of a multipurpose high-temperature oven (able to reach 800 K), heated up by two collar-shaped heaters, one around the reservoir compartment set at 468 K and the other one around the nozzle assembly, slightly warmer at 490 K to maintain a temperature gradient and thus avoid nozzle clogging. The resulting vapor was seeded into 0.5 bar of He and expanded via a 70 μm nozzle to form a molecular beam, which was then collimated by two skimmers (1 and 2 mm) separated by a differential pumping stage[78], before entering the interaction region of the DELICIOUS3 double imaging Photoelectron/Photoion coincidence (i²PEPICO) spectrometer where it crossed the VUV synchrotron radiation (SR) beam from the DESIRS beamline at a right angle. In the following, this method will be referred to as RH.

**Aerosol thermodesorption vaporization method (TD).** Intact gas phase enantiopure parent neutral Pro have been produced in situ from the corresponding homochiral Pro aerosol by thermodesorption on a hot tip inserted in the ionization region, following a method[73] already used for Ala. These aerosols were produced by nebulization of a 1 g L$^{-1}$ solution of Pro in He (2 bars) within an atomizer (TSI, model 3062) followed by a drying stage composed of two silica gel columns before being introduced into a new aerodynamic lens (ADL) that transmits and focuses sub-micron nanoparticles (30–300 nm with 100 % efficiency) into the ionization region[79–81]. As compared to the previously used ADL for Ala, the present one offers a much higher nanoparticle throughput, mechanical stability and reproducibility, and operational duty cycle. After traversing two 2-mm skimmers, the aerosols impinged onto a hot tip made of porous tungsten and heated up *via* a heating cartridge so that the tip temperature was set to 415 and 493 K, the two temperatures used for the two sets of data obtained with the TD method, in conditions referred to as $TD_{415}$ and $TD_{493}$. The released plume of intact neutrals was then ionized by the SR at the center of DELICIOUS3.

**Electron/ion spectrometer.** DELICIOUS3 combines a Velocity Map Imaging (VMI) spectrometer on the electron side with a modified Wiley-McLaren 3D momentum imaging mass spectrometer on the ion side operated in a multi-start/

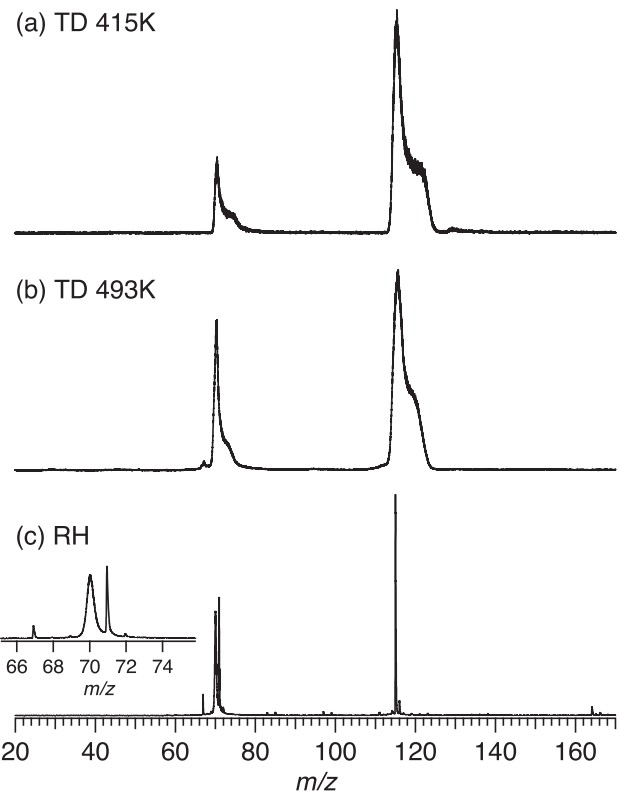

**Fig. 11 Time-of-flight mass spectra (TOF) of proline recorded at 8.7 eV. a** TD$_{415}$ condition. **b** TD$_{493}$ condition. **c** RH condition (ROI filtered) with a zoom in the $m/z$ 65–76 range. The fragments at $m/z$ 67, 69, 71 and 72 are clearly decomposition products in the oven. The asymmetric shapes of the peaks in **a** and **b** are the product of off-axis ions due to the large interaction region and the inhomogeneous extraction field (see ref. [84]).

multi-stop coincidence scheme[82]. This allows performing angle-resolved photo-electron spectroscopy (AR-PES) (with forward/backward capabilities) on mass-selected samples, with possible additional ion imaging/kinetic energy filtering with the RH method. This latter ion imaging filtering (also called region of interest (ROI)) filtering[83] increases the sensitivity of the experiment by selecting ionization events corresponding only to the spatially well-defined supersonic component of the molecular beam, discarding any thermal background contribution. Ultimate electron energy resolution on the detector edge is in the 3% range[78] for the RH method and ~5–10% for the TD method (because of the electrostatic field distortion induced by the TD tip). DELICIOUS3 can also be operated in the threshold photoelectron spectroscopy mode (TPES) with sub-meV up to 10's meV resolution according to an already-described method[84], allowing the study of state-selected cation fragmentation in the so-called TPEPICO scheme[85].

**Characterization of the vapor by mass spectra.** The performance of the RH and TD vaporization methods was checked by observing the ion ToF mass spectra across a range of photon energies (see Supplementary Note 1 and Fig. S4). Both methods produced relatively clean spectra with a strong Pro parent ion ($m/z$ 115) at $h\nu = 8.7$ eV (see Fig. 11), switching to a clearly dominant $m/z = 70$ fragment channel with increasing photon energy. The TD spectra show some asymmetric instrumental peak broadening due to ions from the spatially extended plume of desorbed sample experiencing a greater variability in the non-uniform source extraction field (required for the VMI electron detection). Checks were performed to establish that these broadened peaks are in fact homogeneous (do not encompass varying masses/structures) by examining the electron images (and hence PES and PECD spectra) recorded in coincidence with ions sampled at different points across the peak widths.

On close examination of the RH mass spectra, which is not so afflicted thanks to a tightly collimated molecular beam source, additional much narrower peaks (at $m/z$ 67, 69, 71, and 72) were noted adjacent to the $m/z$ 70 fragment peak. While the latter's width remains consistent with broadening by translational energy release in dissociative ionization, the additional very narrow RH peaks are indicative of non-dissociative ionization from species that have been translationally cooled in the supersonic expansion. We therefore conclude that these narrow peaks evidence that some degree of thermal decomposition occurs in the RH oven prior to reaching the ionization source. These thermal impurities could, however, be

discriminated against by mass filtering, giving confidence that when this is done all three sources yield results attributable to "cleanly" vaporized proline.

**PECD measurements procedure.** For both vaporization methods, PECD for a given enantiomer and photon energy was measured by recording mass-filtered (if needed) and ROI-filtered (for RH only) electron images obtained by alternating CPL helicities, switched every ~15 min. According to a previously detailed procedure[86], the corresponding left- and right-CPL obtained images were merged into two files and then used to provide the PES and the dichroic parameter $b_1$ from, respectively, the total (left + right) and difference (left − right) images after inversion via the pBasex algorithm[87]. The statistical error bars on the dichroic parameter $b_1$ are given as the standard error on the principle that each image pixel acts as an independent counter that follows a Poisson distribution, with the associated error properly propagated through all subsequent operations. We used a single control-energy (8.7 eV) for checking the satisfactory chiral mirroring by repeating measurements with the L- and D-enantiomers; at all other photon energies, PECD was investigated using a single enantiomer, the selection of which was alternated between steps in photon energy in order to cover a greater total range and so optimize the beamtime use.

**VUV photon source.** VUV photons with quasi-perfect circular polarization, above 97% absolute circular polarization rate at the sample location as measured with a dedicated home-made polarimeter[88], were provided by the variable-polarization undulator-based beamline DESIRS[89]. This beamline is equipped with a gas filter suppressing high harmonics of the undulator that could be transmitted by the grating's high orders. The harmonic-free radiation was monochromatized with a 6.65 m-long normal incidence monochromator from which we chose to use the 200 grooves/mm grating providing high flux (in the $10^{12}$–$10^{13}$ photon s$^{-1}$ range) and moderate resolution (typically in the few meV to 10's meV range).

**Numerical methods.** Geometrical coordinates for the four neutral conformers of L-proline were obtained by Møller-Plesset (MP2) and density functional theory (DFT) calculations, made with the B3LYP functional, both using the aug-cc-pVTZ basis set. Electric dipole photoionization matrix elements were subsequently calculated at the DFT geometries by static-exchange calculations using the continuum multiple scattering method[90] with a Slater Xα exchange potential (CMS-Xα)[91,92] following procedures as previously described[43,93–95]. In this study the Xα model potential was constructed using overlapping spherical regions placed at the atomic centers. Radii of these atomic spherical regions were estimated by the Norman algorithm[96] empirically scaled by a factor 0.85. A spherical harmonic angular basis truncated at $l_{max} = 18, 7, 6$ (referring, respectively, to the asymptotic region, the first row atomic regions, and the H atomic regions) was employed for the continuum electron calculations. For the initial neutral state a smaller basis ($l_{max} = 7, 4, 3$) sufficed, while retaining sufficient flexibility to model H atom polarization in the H-bonding interactions[36]. Partial allowance for relaxation effects in the photoionized core were made by using experimentally estimated ionization energies rather than Koopmans approximations in calculating the dipole matrix elements. Subsequently, photo-ionization cross-sections, chiral asymmetry parameters, $b_1$, and PECD were obtained in an independent electron, fixed nuclear geometry approximation using these CMS-Xα photoionization dipole matrix elements.

A further set of neutral molecule calculations were made for thermochemically accurate conformer data by using the G3 composite method[97] and these were repeated for the electronic ground state cation structures. Adiabatic ionization energies were thus estimated by a $\Delta E$ approach using zero-point energy corrected, 0 K energies for each conformer. Vertical ionization energy estimates were calculated using the outer valence Green's function (OVGF) method[98,99] with a cc-pVTZ basis.

Our calculated energies and conformer structures are summarized in Table 1, where they can also be compared with literature values where these are available.

## Data availability
The data that support the findings of this study are available from the corresponding author upon reasonable request.

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

## Acknowledgements

This research was undertaken as part of the ASPIRE Innovative Training Network, which has received funding from the European Union's Horizon 2020 research and innovation program under the Marie Sklodowska-Curie Grant Agreement No. 674960. R.H. and H.G. acknowledge ESR fellowships provided by ASPIRE. We acknowledge the provision of beamtime by Synchrotron SOLEIL (beamtime Proposal 20160125) and we thank the technical staff at SOLEIL for their support and for the smooth operation of the facility, as well as J.-F. Gil for his technical help around the SAPHIRS experiment. We are grateful for access to the University of Nottingham High Performance Computing Facility in support of the computational effort.

## Author contributions

L.N. designed the research project. D.B. conceived the aerodynamic lens. R.H., D.B., H.G., G.A.G., I.P., and L.N. conducted the experiment. R.H. treated the data. G.A.G. performed the internal energy modeling of the breakdown diagrams. I.P. and R.H. performed the electronic structure and PECD theoretical modeling. L.N., R.H., G.A.G., and I.P. wrote the manuscript with contributions from all authors.

## Competing interests

The authors declare no competing interests.
