## [Peer Review File · Communications Chemistry]

Reviewers' comments:

Reviewer #1 (Remarks to the Author):

Hadidi et al. report on a conformer-dependent VUV photodynamics study of gas-phase proline using the DESIRS beamline at SOLEIL. As shown before, proline can exist as at least four different conformers in a molecular beam, and the authors can use the different ionization energies of two groups of conformers to even accomplish conformer-selective PECD. They use their results to also add some speculation about an interstellar origin, driven by PECD, for the homochirality of life on earth.

This is a very dense paper, addressing the VUV photodynamics and enantiomer-selective studies of gas-phase proline in great detail, also considering different temperature regimes. The experimental methods and analysis are certainly state-of-the-art, and new findings on the conformational space of proline are reported. As such, this paper should certainly be published. The current form, however, needs some streamlining, in my opinion, to be better accessible by non-specialists in photoelectron spectroscopy and PECD, which shall be addressed by Communications Chemistry. This is particularly true for the section where the photoelectron spectroscopy results (Figures 3 and 4) are described. In general, it would help the reader if for each figure of the manuscript, it is not only written in the caption WHAT is displayed, but also what can be learned from it. Some problems in understandability might also occur from the special format, i.e., that the experimental details are given at the end. That means, however, that abbreviations (such as RH and TD) need to be introduced upon its first mentioning in the main manuscript (which is currently not the case), otherwise it is tedious for the reader to follow.

While I appreciate that this is the first enantiomer-selective PECD study, I understand that this is also due to the clearly different IEs for the two different conformational classes. The limits in conformer-selectivity following this approach should thus be discussed in a bit more detail: how large does this difference in IEs have to be in order to achieve enantiomer-selective PECD (with the current setup)? Are there other limitations? Otherwise, the impression is left that this is a lucky case.

The authors also speculate on a mechanism to generate ee in interstellar space and how this can be driven by PECD, finally being responsible for homochirality of life. While I think that this is a valid speculation, the speculative character should probably be made a bit clearer. There are many open questions that would become relevant in such a scenario: What if the opposite handedness would have brought to early earth via a different event as well? How many molecules of one handedness are we talking about? How much spatial separation would have been needed in order to pick up only one handedness? What could PECD achieve under realistic conditions?

The paper also contains some editing issues, such as too many spaces within a sentences here and there as well as half sentences being started by a comma, but not properly closed. While streamlining the paper for better readability for non-specialists, the authors should thus also go through it and remove such issues.

Reviewer #2 (Remarks to the Author):

In this manuscript, Hadidi et al. have used TPES/TPEPICO and PECD techniques, complemented by ab initio calculations and theoretical fitting, to present a comprehensive photodynamic study of proline. The authors first reviewed and calculated the energetics and structural parameters of the four most commonly studied proline conformers and measured the PES with respect to three thermal deposition conditions. The different PES profiles of these conditions are carefully analyzed, and the explanation is consistent with experimental findings. HOMOs of each conformer were calculated to illustrate their correlation with the IE for each conformer. In the second part, the authors studied the PECD of each system combined with theoretical fitting. The content of this manuscript is well-structured, and relevant literature has been extensively reviewed which provided a solid background introduction. The conclusion, although a little lengthy, is strongly supported by the experimental findings and theoretical analyses. Before it gets accepted, I have only several minor questions regarding the photoelectron spectroscopy part from my expertise.

1. In order to enhance the overall legibility, I would recommend the authors:

- a) As the "Methods" section comes after the main text, the full name of many abbreviations that appear in the Methods section should be mentioned in the main text.
- b) Since Fig. 1 and Fig. 2 are not very informative, I think these two can be combined for an easier way of presentation.
- c) Adding x-axis for all the three frames in Fig. 3.
- d) The labeling of figures in Supporting Information is problematic and does not correlate to the main text.

2. For the Methods section, could the author justify why the two temperatures (415K and 493K) were chosen? Was thermal decomposition observed at higher temperatures? From Fig. S1 (or should be S4?) I notice that higher temperature favors more fragmentation, I may doubt that if $m/z = 70$ has any contribution from thermal decomposition.

3. In Fig. 3, the lower frame, $m/z = 115$ exhibits significant noise/uncertainty, and the threshold are not very clear. Regardless of thermal decomposition causing weak overall signal intensity, can the authors add some details about the acquisition of each spectrum such as scan time, and comment on this?

4. I quite agree with the author that the TD method generated a much colder molecular beam compared with RH. Did the authors observe any molecular clusters? I am curious that if the profile of PES was influenced by weakly bounded van der Waals clusters.

5. The proline mentioned throughout the first half of the manuscript, before PECD, is L-proline only?

Reviewer #3 (Remarks to the Author):

The authors report on conformer dependent photodynamics in the VUV regime of gas phase proline.

Applying photoelectron/photoion coincidences proline was investigated. Here the authors make use of the fact that the measurement of the photoelectron acts as a tag for the conformer.

Dependencies of the photodynamics, as state-selected ion fragmentation, and photoelectron circular dichroism, are reported. The experimental work and the accompanying theory seems solid

and carried out carefully. The results are convincing and extensively discussed. Besides a few minor comments, I have only one major criticism:

Are the results novel? Yes, the results are novel, but only for this molecule (out of thousand others).

Will the results be of interest to others in the community and the wider field? Within the community, the results are generally interesting, for a wider field, I don't think so. The results are similar interesting, new and "unexpected" as total photo absorption cross sections of any other "bio"-molecule. I haven't seen any, spectacularly unexpected results/knowledge gain, which deserves publication here. Or I overlooked this point. And I'm happy to change my opinion, if the authors have convincing answers. A more specialized journal (J. Phys. Chem. or so) seems at this moment to be more appropriate.

Minor points:

The abbreviations TPES and TPEPICO are not written out in words; TD and RH are also not explained when used the first times.

It would be nice to read earlier in the manuscript that the conformer concentration/appearance is actively manipulated and not a natural mix.

Error bars in Fig. 3 are missing or at least a comment in the figure description (is the spiky structure real or an artefact).

Why is in Fig. 6 D-Pro with TD_493 or L-PRO with TD_415 not presented? It would be nice to see that the other enantiomer shows the inverted signal.

In line 399 it probably should be TD_493 (instead of TD_495).

It is not clear to me, what this experiment adds to the discussion of the homochirality of life. Or in other words, what is purpose of the extended discussion "Astrochemical implications" in the context of this paper?

Also in Fig. 11 are the error bars missing.

We are grateful to the 3 reviewers for their helpful comments and queries which helped us to improve the manuscript. Below are our answers (in italic) to each of the Reviewers' comments:

Reviewer #1 (Remarks to the Author):

Hadidi et al. report on a conformer-dependent VUV photodynamics study of gas-phase proline using the DESIRS beamline at SOLEIL. As shown before, proline can exist as at least four different conformers in a molecular beam, and the authors can use the different ionization energies of two groups of conformers to even accomplish conformer-selective PECD. They use their results to also add some speculation about an interstellar origin, driven by PECD, for the homochirality of life on earth.

This is a very dense paper, addressing the VUV photodynamics and enantiomer-selective studies of gas-phase proline in great detail, also considering different temperature regimes. The experimental methods and analysis are certainly state-of-the-art, and new findings on the conformational space of proline are reported. As such, this paper should certainly be published. The current form, however, needs some streamlining, in my opinion, to be better accessible by non-specialists in photoelectron spectroscopy and PECD, which shall be addressed by Communications Chemistry. This is particularly true for the section where the photoelectron spectroscopy results (Figures 3 and 4) are described. In general, it would help the reader if for each figure of the manuscript, it is not only written in the caption WHAT is displayed, but also what can be learned from it.

Authors answer: *In order to make the manuscript more understandable by a very broad audience, and according to the reviewer's suggestion, we added:*

- *at the end of the (former) Fig. 3 (now Fig.2) caption the two sentences:
"While the upper panel shows the full TPES, obtained by recording all threshold electrons – those having approximately zero kinetic energy such that all photon energy in excess of the ionization limit is deposited as internal excitation of the cation – the lower panels only show those threshold electrons that are found by TPEPICO coincidence detection to be accompanied by production of either parent or fragment ions. Clear temperature effects are observed on the shape and onsets (see text) of these mass-selected TPES."*
- *At the end of the (former) Fig.4 (now Fig.3) caption, the two sentences:
"Type I conformer orbitals appear very localized at the N atom site (coloured blue) but this is much less the case of conformer II orbitals showing more delocalization with some density also at the O atom sites (red) and, notably, around the C- α bond. These differences may partially account for the different fragmentation behavior between the two types of conformer".*

And we changed the caption of the (former) Fig.5 (now Fig.4) as:

"Fig. 4. PES (solid curves) and PECD (discrete symbols with error bars) obtained with D- (blue) and L- (red) proline enantiomers, produced in TD₄₁₅ conditions, and extracted from a given pair of electron images recorded with left- and right- circularly polarized light (CPL) using photon energies of (a) 8.7 eV; (b) 9.5 eV; (c) 11.5 eV. The images include electrons coincident with formation of all masses associated with the ionization of Pro (m/z 115 + m/z 70). PES profiles can be understood as relative cross-sections for ion formation and at a given ionization energy will correspond to formation of a specific excited state of the ion; however, as photon energy is increased the kinetic energy for electrons produced for a given ionization energy (state) will be likewise increased. A clear dependence of b_1 with both the ion state and the associated electron kinetic energy is thus observed. Note the very satisfactory expected L/D mirroring in panel (a). Inserts show raw difference photoelectron velocity map images obtained by subtracting left- and right-CPL measurements. The images show a forward/backward asymmetry with respect to the CPL propagation axis."

Note that former Fig.5 (now Fig.4) has been improved with additional graphical information regarding the electron raw image differences.

Some problems in understandability might also occur from the special format, i.e., that the experimental details are given at the

end. That means, however, that abbreviations (such as RH and TD) need to be introduced upon its first mentioning in the main manuscript (which is currently not the case), otherwise it is tedious for the reader to follow.

Authors answer: *The reviewer is right and we understand perfectly this criticism which is linked to the format of the journal with the methods section placed at the very end of the manuscript. We have asked the editor if the methods section could be moved earlier in the text, but this turned out to be impossible. However, the editor suggested that we add a “methods” paragraph at the beginning of the results section. We decided to follow this suggestion by adding such a paragraph providing some background on experimental methods and defining acronyms, so that the reading of the paper gets smoother. Therefore, in page 3, at the very beginning of the “results and discussion” section we added the following short paragraph:*

“We used two complementary vaporization methods to bring Pro into the gas phase: Resistive Heating (RH) associated to an adiabatic expansion and aerosol ThermoDesorption (TD) at two different temperatures, which leads to the 3 vaporization conditions of the neutral labelled as RH, TD₄₁₅, TD₄₉₃ respectively. After ionization of these neutrals by VUV synchrotron radiation (SR), the corresponding electrons and ions are detected in coincidence by imaging (time and position sensitive) detectors. By considering only threshold electrons (i.e. those with zero kinetic energy), while scanning the photon energy, one may retrieve the total Threshold PhotoElectron Spectrum (TPES) by analysis of the electron image. By selecting only those electrons coincident with a chosen ion mass the so-called Threshold PhotoElectron/Photolon COincidence (TPEPICO) spectrum provides a state-selected fragmentation pattern. Considering instead all (i.e. fast and slow) electrons collected at a fixed photon energy, our set-up provides, from the radial and angular distribution observed in a (mass-filtered) electron image, the angle resolved PhotoElectron Spectrum (PES). From pairs of images recorded with left- and right- circularly polarized light the chiral angular parameter, b_1 , can be extracted as a function of electron kinetic energy. (for more details see the Methods section)”

While I appreciate that this is the first enantiomer-selective PECD study, I understand that this is also due to the clearly different IEs for the two different conformational classes. The limits in conformer-selectivity following this approach should thus be discussed in a bit more detail: how large does this difference in IEs have to be in order to achieve enantiomer-selective PECD (with the current setup)? Are there other limitations? Otherwise, the impression is left that this is a lucky case.

Authors answer: *Besides the fundamental motivations for studying Proline by itself (see introduction), we took advantage of the large (~0.8 eV) IE difference between the two types of conformers offered by the proline showcase, to tag them. Such a large energy difference is not very common, and for most amino-acids for instance, in general the two conformer types have lower IE separation (0.1- 0.3 eV see Ref. 17) but some others (aspartic acid, asparagine, methionine, lysine, arginine and tryptophan) have larger type I / type II IE separations, in the 0.3-0.6 eV range, which is probably the lower limit range for which our set-up/method can be used. Even if the presence of sub-class conformers (like C and D for conformer type I) have to be considered, as well as spectral congestion, for some of these amino-acids the method could therefore be generalized.*

We therefore added a few lines in the first sentence of the before last paragraph of the conclusions, which reads now:

“As an extension of this first chiroptical study of proline, and besides possible applications of our conformer-tagging method to the few other amino-acids possessing quite large Type I/ Type II conformer IE separation (~0.3-0.6 eV) [ref 17], future experimental efforts for a direct study of conformer-specific PECD on floppy systems could take two avenues:...”

However, our main claim on conformer-dependent PECD is not focused on the generalization of the method, but rather about the first ever observation of conformer-dependent PECD. Indeed, there is solid theoretical evidence that conformers have markedly different PECD; hence PECD offers an excellent chance to probe conformation. However, to date PECD has experimentally dealt with conformer mixtures, and these cannot always be assumed to be in thermal equilibrium. Attempts to deconvolute and study individual conformers have thus been rendered inconclusive due to the

uncertain relative conformer populations. Proline (and a few other amino-acids as listed above) with its very large conformer type IE separation offers therefore a perfect showcase to directly demonstrate experimentally the conformer-specificity of PECD.

This statement is crucial in the context of laser-based PECD application in analytical chemistry and probably soon in analytical biochemistry. Indeed, resonance-enhanced multi-photon ionization (REMPI) schemes are able to fully resolve conformers in a mixture (see Ref. 35). Recent demonstration of REMPI-PECD with a nanosecond pulsed laser now clearly shows how a PECD measurement can be combined with a high resolution (see Ref. 36). In such a scheme, a conformer-selective excitation to achieve a fully conformer selective measurement would not necessarily require large gaps in the conformer IE. The present work thus provides a clear indication how PECD, with appropriate excitation/ionization methods can be successfully applied to study conformer structures in a wide range of chiral molecular systems.

This latter point has been highlighted in the conclusions adding the following sentence at the end of the before last paragraph of the conclusions:

“Conversely, while having been predicted by theory as soon as year 2000 (ref. 74), this first ever experimental evidence of conformer-dependent PECD is important, considering the quickly growing field of table-top laser-based REMPI-PECD for in situ ee measurement in analytical chemistry (ref. 75 & 76) and probably soon analytical biochemistry for which conformational flexibility will be a challenging issue to deal with”.

The authors also speculate on a mechanism to generate ee in interstellar space and how this can be driven by PECD, finally being responsible for homochirality of life. While I think that this is a valid speculation, the speculative character should probably be made a bit clearer. There are many open questions that would become relevant in such a scenario: What if the opposite handedness would have brought to early earth via a different event as well? How many molecules of one handedness are we talking about? How much spatial separation would have been needed in order to pick up only one handedness? What could PECD achieve under realistic conditions?

Authors answer: *We thank the reviewer for his/her interest in the astrochemical aspect of our work. We provide hereafter some short answers but feel that a full development of those in the manuscript would not be appropriate for length considerations and also because some background on our astrochemical scenario has already been published (see for instance ref. 29,33, 34). Briefly:*

What if the opposite handedness would have brought to early earth via a different event as well?

Authors answer: *The origin of life's homochirality has been the subject of numerous scenarios trying to rationalize the production of a sizable ee. To our knowledge the competition of several scenario has never been considered and even less quantified. One could imagine that the handedness with the highest ee would win a possible “handedness war”, considering that homochirality is a chance and an asset for the development of life, a development which appears unlikely in a racemic world (see ref. 47)*

How many molecules of one handedness are we talking about?

Authors answer: *Absolute numbers are meaningless since nobody knows how much organic material life started from. As it is the case of all alternative scenario, we are only considering relative figures. At the Lyman- α wavelength, the parent survival is in the few % range (see fig.11 (now Fig.10)), with ee varying from ~ 12 % to ~ 3% depending on temperature (see Fig. 10 (now Fig. 9)). This provides macroscopic numbers, well above statistical sampling uncertainties, which is not the case for instance of parity-violating based scenario.*

How much spatial separation would have been needed in order to pick up only one handedness?

Authors answer: *From Fig. 3, the parent peak, at the lowest temperature, peaks at ~8.8 eV binding energy, corresponding therefore to 1.4 eV electron kinetic energy at the Lyman- α photon energy (10.2 eV). Because of momentum conservation, this corresponds to a parent (m/z 115) ion recoil speed of ~*

$3.5 \text{ m}\cdot\text{sec}^{-1}$. At this speed, only 1.5 million years are required to travel one au (astronomical unit = distance from earth to sun). This back-of-an-envelope calculation simply shows that over astronomical times (for which a million year is nothing) the two enantio-enriched clouds would have been much more separated than the size of the proto-solar system.

What could PECD achieve under realistic conditions?

Authors answer: In practice, as for all the alternative CPL-based scenarios, the achieved ee should be multiplied by the actual absolute circular polarization rate of the VUV light encountered in the ISM, which typically lies in the few % up to 20 % (see ref. 58). Therefore, the realistic achievable PECD-induced ee lies in the 10^{-3} - 10^{-2} range which is still higher than scenarios based on magneto-chiral effects for instance in realistic ISM conditions. In order to elaborate on this, in the manuscript, we added a sentence at the end of the astrochemical implications section, in page 20:

“Note that in realistic ISM conditions, as for all the other CPL-based scenarios, these ee figures should be multiplied by the actual absolute circular polarization rate of the VUV radiation encountered in heavy star forming regions, typically lying in the few % to 20 % range [ref 58], leading to an effective ee in the 0.1 % to 2 % range.”

The paper also contains some editing issues, such as too many spaces within a sentences here and there as well as half sentences being started by a comma, but not properly closed. While streamlining the paper for better readability for non-specialists, the authors should thus also go through it and remove such issues.

Authors answer: About twenty double spaces have been suppressed in the revised manuscript (they were due to mac/windows MS-word mismatches).

Reviewer #2 (Remarks to the Author):

In this manuscript, Hadidi et al. have used TPES/TPEPICO and PECD techniques, complemented by ab initio calculations and theoretical fitting, to present a comprehensive photodynamic study of proline. The authors first reviewed and calculated the energetics and structural parameters of the four most commonly studied proline conformers and measured the PES with respect to three thermal deposition conditions. The different PES profiles of these conditions are carefully analyzed, and the explanation is consistent with experimental findings. HOMOs of each conformer were calculated to illustrate their correlation with the IE for each conformer. In the second part, the authors studied the PECD of each system combined with theoretical fitting. The content of this manuscript is well-structured, and relevant literature has been extensively reviewed which provided a solid background introduction. The conclusion, although a little lengthy, is strongly supported by the experimental findings and theoretical analyses. Before it gets accepted, I have only several minor questions regarding the photoelectron spectroscopy part from my expertise.

1. In order to enhance the overall legibility, I would recommend the authors:

a) As the “Methods” section comes after the main text, the full name of many abbreviations that appear in the Methods section should be mentioned in the main text.

Authors answer: The reviewer is right and we fixed this issue by adding a short paragraph in the revised manuscript at the very beginning of the results section (see our response the second comment of reviewer 1).

b) Since Fig. 1 and Fig. 2 are not very informative, I think these two can be combined for an easier way of presentation.

Authors answer: Yes this is a good suggestion. Therefore, Fig 1 & 2 have been merged into a new Fig.1 (as well as their caption), and all the other figures have been renumbered.

c) Adding x-axis for all the three frames in Fig. 3.

Authors answer: We understand the point but for the sake of clarity, we prefer to leave Fig. 3 as it is, since it's obvious that the 3 panels share the same x-axis.

d) The labeling of figures in Supporting Information is problematic and does not correlate to the main text.

Authors answer: *The reviewer is right, sorry for this editing mistake and thanks to the reviewer for spotting it. Indeed, S1 should be labeled S4, S2 should be S1, S3 should be S2 and S4 should be S3. We corrected this mistake in the revised supporting information.*

2. For the Methods section, could the author justify why the two temperatures (415K and 493K) were chosen? Was thermal decomposition observed at higher temperatures? From Fig. S1 (or should be S4?) I notice that higher temperature favors more fragmentation, I may doubt that if $m/z = 70$ has any contribution from thermal decomposition.

Authors answer: *The main motivation was to obtain a set of temperatures with the widest range in order to achieve the largest changes in the conformer population distribution. 415 K at the tip of the thermodesorber was the lowest temperature leading to a sufficient density of the neutral plume, while 493 K was the largest temperature we could achieve considering the limited power of the thermal cartridge and thermal losses induced by the necessary large distance between the cartridge and TD tip, and by radiation. So, we could not go further in temperature and therefore could not observe any thermal decomposition if any would occur.*

Regarding Fig S4 (former S1), an increase of temperature favors the fragmentation of the cation in a hot ground state (a very well documented effect of dissociative photoionization, see Ref. 71 and 72), not a thermal decomposition of the neutral (which we never observed with the TD method for any amino-acids). Would the latter process occur, the m/z 70 peak in the RH panels (c) would have a double structure: a sharp peak (for neutral decomposition) superimposed onto a broad one corresponding to the genuine dissociative ionization of intact neutral proline (KER effect). This is not the case. Also, by filtering the electron image with various part of this peak (center or wings) we find the same associated PECD, meaning that we are sampling the same neutral molecule: the intact parent Pro.

3. In Fig. 3, the lower frame, $m/z = 115$ exhibits significant noise/uncertainty, and the threshold are not very clear. Regardless of thermal decomposition causing weak overall signal intensity, can the authors add some details about the acquisition of each spectrum such as scan time, and comment on this?

Authors answer: *Thermal decomposition is not really the issue here but rather the typical low ionization cross section in the threshold region, associated to a low target density. These are difficult experiment and we had to accumulate the signal for acquisition time ranging from 85 sec/points (TD_{493}) to 160 sec/points (RH). This rather poor Signal-to-noise ratio makes that our threshold determination possesses a limited accuracy (see table 2 last column), getting worse for the RH method because of its corresponding limited neutral density.*

In the caption of Fig.3 (now Fig.2) we added "for acquisition time ranging from 85 sec/points (TD_{493}) to 160 sec/points (RH)"

4. I quite agree with the author that the TD method generated a much colder molecular beam compared with RH. Did the authors observe any molecular clusters? I am curious that if the profile of PES was influenced by weakly bounded van der Waals clusters.

Authors answer: *It is true that the presence of clusters would affect the PES (and the PECD profile in fact). This is a well-documented effect (see ref 28 for a collection of references on clustering effect on PECD). The usual effect on the PES is a red-shifting on the parent mass (due to cascading dissociation ionization processes) which we did not observe, nor in any trace of stable dimer at m/z 230. This is not surprising at all since even with the TD_{493} method (providing a plume of neutral not a molecular beam per se) the internal temperature was determined at 384 K (see Table 2), a temperature at which no Van der Waals nor H-bonded clusters can be formed and survive (this would require a quite strong adiabatic expansion leading to much colder temperatures to be formed, see for instance Nahon et al. Phys. Rev. A **82**, 032514 (2010) or Powis et al., PCCP **16**, 467 (2014)).*

5. The proline mentioned throughout the first half of the manuscript, before PECD, is L-proline only?

Authors answer: *Before the PECD section starts in page 10, the enantiomeric content of the sample doesn't matter since PES and fragmentation are not enantio-dependent this is why we didn't specify the enantiomer. But yes, for the spectroscopic part (Fig.3 and 4) we used the natural (and cheaper) L-proline.*

Reviewer #3 (Remarks to the Author):

The authors report on conformer dependent photodynamics in the VUV regime of gas phase proline.

Applying photoelectron/photoion coincidences proline was investigated. Here the authors make use of the fact that the measurement of the photoelectron acts as a tag for the conformer. Dependencies of the photodynamics, as state-selected ion fragmentation, and photoelectron circular dichroism, are reported. The experimental work and the accompanying theory seems solid and carried out carefully. The results are convincing and extensively discussed. Besides a few minor comments, I have only one major criticism:

Are the results novel? Yes, the results are novel, but only for this molecule (out of thousand others).

Authors answer: *Proline is a major proteic amino-acid (out of 22) with very specific structural properties. New information regarding its electronic and molecular structures (as well as fragmentation/stability), in a controlled environment where solvent effects are avoided and high level ab initio methods can be applied, is therefore important since it will have an impact in the ability of proline to be involved in the building-up of peptides and larger biopolymers. This is why the sole proline molecule has been the subject of previous papers published in general chemistry journals such as Angew. Chem. (see ref. 8 and 15).*

*Besides, we studied also proline as a benchmark molecule (enabled by the large IE separation between conformer types) to demonstrate for the first time ever the possibility to measure conformer-dependent PECD which is probably the most challenging and stringent test for PECD modeling. Conformer-dependent PECD was predicted 20 years ago (see Powis, I. J. Phys. Chem. A **104**, 878, (2000)) but never measured directly prior to this study. In this sense, the outcome of our work goes far beyond the case of proline, especially in the context of the growing field of laser table-top REMPI-PECD experiment with a strong chemistry analytical application and soon very likely bio-chemical applications for which conformer-related issues will be of major relevance. We strengthened this point in the conclusion of the revised manuscript (see our response to the third comment of reviewer 1).*

Will the results be of interest to others in the community and the wider field?

Authors answer: *Yes we believe that the outcome of our study will be in of interest to a broad range of chemistry-related communities of chemical/molecular physics (fundamental insights into photoionization dynamics), physical chemistry (fragmentation/conformational landscape), analytical biochemistry and mass spectrometry, organic chemistry (chirality) and astrochemistry/origin of life.*

Within the community, the results are generally interesting, for a wider field, I don't think so. The results are similar interesting, new and "unexpected" as total photo absorption cross sections of any other "bio"-molecule. I haven't seen any, spectacularly unexpected results/knowledge gain, which deserves publication here. Or I overlooked this point. And I'm happy to change my opinion, if the authors have convincing answers. A more specialized journal (J. Phys. Chem. or so) seems at this moment to be more appropriate.

Authors answer: *In addition to the response above (as well as to reviewer 1's third comment), we want to stress the specific sensitivity to molecular structures of PES and most of all PECD as compared to total absorption cross sections. Likewise, the chiral sensitivity of PECD exceeds that of essentially all other chiroptical methods, such as CD, by orders of magnitude. For example, in Figs 4 & 5 (formerly 5 & 6) we report chiral asymmetries of 10% —20%. This is so because PECD is observed via a differential angular measurement for which the pure electric dipole (E1) term is non-zero, while*

total ionization cross sections (or CD in absorption) are measured via an integral measurement for which, by symmetry considerations, this E1 term vanishes (see ref 26 and for instance ref 28 for a review). To our knowledge there are no UV absorption spectra of amino acids able to decipher conformers, while our PECD study is conformer-dependent (again for the first time) and our experimental / theory interplay allowed to refine the conformational landscape of Pro. This is a major point since in an analytical context of ee determination, temperature effects, and therefore conformational effects, should be taken into account.

Additionally, the Pro cation conformer-dependent fragmentation behavior that we unraveled is unexpected (and quite unique).

Finally, the confirmation of the astrochemical scenario suggested for alanine is, we believe, an important result (see response below).

All our findings are wrapped up in the conclusion (with the revisions mentioned above in the answers to reviewer 1's 3rd comment) and we believe they are important and relevant to a wide audience, demonstrating for the first time conformer-specific PECD and providing a complete and self-consistent overview of proline VUV photodynamics applicable to various laboratory contexts as well natural environments such as the ISM.

Minor points:

The abbreviations TPES and TPEPICO are not written out in words; TD and RH are also not explained when used the first times.

Authors answer: The reviewer is right and we fixed this issue by adding a short paragraph in the revised manuscript at the very beginning of the results section (see our response the second comment of reviewer 1)

It would be nice to read earlier in the manuscript that the conformer concentration/appearance is actively manipulated and not a natural mix.

Authors answer: We are not manipulating the conformer by any means. They are populated according to their stability ordering (assuming a thermal equilibrium in our experimental conditions), the only "manipulation" is to vary the temperature to vary the conformer population distribution. This is mentioned quite early in the text, i.e. in the introduction line 107-110. However, the conformer mixture is "natural", although not at room temperature because of the vanishing vapor pressure of proline at room temperature.

Error bars in Fig. 3 are missing or at least a comment in the figure description (is the spiky structure real or an artefact).

Authors answer: As written in the caption of Fig. 2 (formerly 3), the error bars are indicated by the width of the shaded areas around each curve. They increase with the photon energy because of false coincidences (uncorrelated electrons) due to increasing ionization of background species. The spiky structures are basically noise, the limited SNR is caused by the limited neutral density, especially for the RH condition.

Why is in Fig. 6 D-Pro with TD_493 or L-PRO with TD_415 not presented? It would be nice to see that the other enantiomer shows the inverted signal.

Authors answer: In an ideal world, having dual L and D data for each photon energy would be great. In practice, these experiments are time consuming and synchrotron beamtime is not infinite. We chose to acquire a set of data for each enantiomer at different photon energy, to cover a wide VUV range, with a single control photon energy at 8.7 eV (see Fig.5(a)) in order to check and show the nice L/D mirroring of the data. This is a common practice in most of the PECD literature.

In line 399 it probably should be TD_493 (instead of TD_495).

Authors answer: The reviewer is right. This typo has been corrected in the revised version. We corrected as well exactly the same typo in Fig. 9.

It is not clear to me, what this experiment adds to the discussion of the homochirality of life. Or in other words, what is purpose of the extended discussion "Astrochemical implications" in the context of this paper?

Authors answer: *We thank the reviewer for offering the possibility to stress the relevance of this part of our work which we think is important as demonstrated by the genuine interest shown by reviewer 1.*

The astrochemical implications section describes a direct outcome of our PECD study, in link to an astrophysical PECD-induced possible origin of homochirality as we originally suggested via a study of alanine (ref 33 & 34). Pro is of major astrobiological interest (one of the first amino-acids recruited by life) and it is therefore important to make sure that the PECD-based scenario suggested for Alanine holds for Pro. It is the case since the PECD at the Lyman- α photon energy has the same sign for both Ala and Pro. Moreover, because in the case of Ala, the Lyman- α PECD happens by chance to be the same for the 3 main conformers, it was not possible to get any temperature information in terms of sign/magnitude of PECD (the Boltzmann-averaged PECD was insensitive to the conformer distribution, i.e. to the temperature). Here, via our thorough experiment/theory PECD interplay on Pro, we are able to state that our scenario is valid whatever the temperature (with, in particular, a constant sign, the same as the sign of alanine), i.e. whatever the astrophysical environment. This is a major point, in direct connection with the whole study (involving also the parent survival aspect which investigation required our PEPICO approach).

Also in Fig. 11 are the error bars missing.

Authors answer: *A thorough error propagation analysis had been carried out, and the corresponding calculated errors are within the symbol size, which we indeed didn't specify in the initial manuscript.*

In the revised manuscript, we added at the end of the caption of (former) Fig. 11 (now Fig. 10): "The statistical error bars (standard deviation) are within the symbol size".

Reviewers' comments:

Reviewer #1 (Remarks to the Author):

[Editorial Note: This reviewer did not provide any further comments for the authors.]

Reviewer #2 (Remarks to the Author):

The authors have very properly addressed all of my questions. Now I think the manuscript is ready to publish.

Reviewer #3 (Remarks to the Author):

Hadidi et al. have revised their manuscript on gas-phase proline conformer-dependent VUV photodynamics.

After reading the manuscript a second time, I have a few more comments.

The Paper has interesting results to be present, which should be published. However, I still think that these are only of interest for a small group of specialists and not a broader audience.

I still think that the manuscript doesn't have any significant contribution to the field. Neither PECD is a new tool, nor the molecule is new, nor is the gained knowledge new. It is known since a long time that proline has 4 conformers and it is obvious that each of the conformers shows a different PECD, when correctly prepared. And this is the only contribution of this study. The experiment and the results are rather straight forward in this community. Conformer dependent PECD has already been measured 10 years ago (reference 30 and 32), as the authors already stated - on alaninol, which is an amino alcohol, instead of proline (an amino acid). Also the separation of conformers depending on the ionization has been shown in the past.

It is surprising that most of the figures seem to show different amounts of data points, which is questionable. Did the authors removed certain data points, as they didn't fit? Let me give some examples:

Why are in Figure 4 (b) and (c) not the inverted graphs for the other enantiomer shown? Also the comment "very satisfactory in the description of Figure 4 (a) stress this point a bit to much. Both curves here show a similar trend, decreasing towards 0, but it seems that within the error bars a mirroring of the curves would show significant discrepancies, as the blue for example shows an oscillation, which is not present in the red curve. The error bars seems not being real. Why? Rule of thumb is that for a statistical error on average every 3rd point should be displaced out of the error bars, while here the curves are surprisingly smooth. Also the insets in (b) and (c) seem more noisy than the shown PECD is. Or are the insets just a subset?

In Figure 5 are shown the results for $m/z = 115$ (top), 70 (middle) and total (bottom row). Why are in the total data so many points missing? Here are basically the data from $m/z=70$ shown. And even the PES is an exact copy of the middle row (at least for left and middle column).

Why aren't the experimental results from Figure 5 not overlaid with the theory from Figure 6?

In Figure 7, it seems that data points are missing. What happened to the red data point around 2 eV and why are the blue and green points not shown for higher energies, while a fit to experimental data is shown. How can results be fitted, without data points?

Similarly in Figure 8. Where are the data points for green/blue at higher kinetic energies? They have obviously been recorded, as Figure 2 suggests. And in Figure 2 it would be nice to put some numbers on the y-axis - just to better show that the data are shown on a linear scale.

Proline is an amino acid as the authors wrote. However it naturally appears an inner salt / zwitterion. And within a 30 pages manuscript, I miss a discussion on this. It is essential to show that the molecule, which is investigated is safely brought intact into gas phase and not destroyed. The mass specs in Figure 11 (a) and (b) clearly indicate that the hump towards higher masses that dimers/clusters are created, which then fragment -> Thus a larger mass, depending on the additional group is observed. And even the correct mass of a molecule, doesn't prove that the molecule is present in its shape/conformer, as expected. I miss a clear discussion on this point.

The discussion at the end of the manuscript regarding astrophysical relevance / homochirality is a completely detached topic from the paper. The obtained results are may be interesting for the future research here, but such a long discussion of things, which the authors didn't work on, is only nice to read, but not of any relevance.

In conclusion, I suggest to shorten the manuscript, stressing the essential points and submitting it to a more specialized journal.

We are pleased that the first two reviewers are satisfied with the last version of the manuscript appearing to them as suitable for publication in *Commun. Chem.*

Reviewer 3 is, however, still raising concerns, several of which are simply a repetition of issues that have previously been raised. While it is of course open to a reviewer to come back on points where a previous response was not considered satisfactory, we find it unhelpful that in repeating questions here there is no recognition or acknowledgement of our previous responses. We do not then know whether these were ignored or were considered deficient in some way. If the latter we are denied the opportunity to attempt further manuscript changes which might help improve the accessibility of our paper to a wider, non-specialist readership (something we would be happy to do). We do, however, take the opportunity to address and avert some misconceptions and misunderstandings of the experimental methodology that are apparent to us in some of the comments.

Below are our answers (in italic) and revisions (underlined in yellow) to each of Reviewer 3's comments:

Reviewer #3 (Remarks to the Author):

1-On the technical issues

Reviewer 3: It is surprising that most of the figures seem to show different amounts of data points, which is questionable. Did the authors removed certain data points, as they didn't fit? Let me give some examples:

Authors answer: *We note the implication of inappropriate data manipulation and we are happy to be able hereafter to refute and fully address all these concerns.*

Reviewer 3: Why are in Figure 4 (b) and (c) not the inverted graphs for the other enantiomer shown?

Authors answer: *The same reviewer asked the same question in the first evaluation round regarding the then Fig.6 (now Fig. 5). As we responded in the first review:*

*"In an ideal world, having dual L and D data for each photon energy would be great. In practice, these experiments are time consuming and synchrotron beamtime is not infinite. We chose to acquire a set of data for each enantiomer but at different photon energies, to cover a wide VUV range, with a single control photon energy at 8.7 eV (see Fig.4(a)) in order to check and show the nice L/D mirroring of the data. This is a common practice in most of the PECD literature" (see for instance Nahon et al. PCCP **18**, 12696 (2016)).*

However, for the sake of clarity we now add in the Methods section, page 23 & 24:

- *A new subheading "PECD measurements procedure"*
- *A new sentence, at the end of this subsection: "We used a single control-energy (8.7 eV) for checking the satisfactory chiral mirroring by repeating measurements with the L- and D-enantiomers; at all other photon energies, PECD was investigated using a single enantiomer, the selection of which was alternated between steps in photon energy in order to cover a greater total range and so optimize the beamtime use."*

Reviewer 3: Also the comment "very satisfactory in the description of Figure 4 (a) stress this point a bit to much. Both curves here show a similar trend, decreasing towards 0, but it seems that within the error bars a mirroring of the curves would show significant discrepancies, as the blue for example shows an oscillation, which is not present in the red curve..."

Authors answer: *We agree with the reviewer that the mirroring in Fig. 4(a) is slightly outside of the error bars. This is because the error bars take into account only statistical errors, as explained at the bottom of page 23 in the Methods section. There are probably some additional limited systematical*

errors, such as imperfect and different enantiopurity of the samples, or different S/N for both enantiomers due to the preparation of different solutions to be nebulized, which may lead to a slightly non-zero baseline between the two enantiomers as often encountered in CD experiments. We could have normalized the data of Fig. 4a, as often done in absorption CD works, so that the mirroring would be better, but we did not, nor did we smooth the data.

In order to take into account the remark of the reviewer, in the caption of Fig.4, we changed “the very satisfactory expected L/D mirroring” into “the satisfactory and expected L/D mirroring”.

Reviewer 3:The error bars seems not being real. Why? Rule of thumb is that for a statistical error on average every 3rd point should be displaced out of the error bars, while here the curves are surprisingly smooth.

Authors answer:

Our derivation of error bars by standard statistical treatments is properly described in the Methods section (p. 23):

“The statistical error bars on the dichroic parameter b_1 are given as the standard error on the principle that each image pixel acts as an independent counter that follows a Poisson distribution, with the associated error properly propagated through all subsequent operations.”

This is a widely recognized, statistically based approach to data treatment, whose principles and formulae are fully documented in many textbooks. An arbitrary variation of the error bars to satisfy some undocumented rule of thumb would not be appropriate. We do, however, clearly state at points in the manuscript that these statistical error bars exclude any systematic error.

Reviewer 3: Also the insets in (b) and (c) seem more noisy than the shown PECD is. Or are the insets just a subset?

Authors answer: *The insets in Fig.4 (b) and (c) are raw difference images from which, after Abel inversion, the b_1 curve is obtained. The electron counts in the ~65000 pixels in an image are effectively combined and reduced by the data treatment to a much smaller small number of $b_1(E)$ parameters. The statistical (Poisson) noise seen in individual pixels is thus quantitatively transformed in the data reduction to the uncertainty of those smaller number of parameters, as described in the previous response. Hence, an intuitive visual comparison of the “pixel noise” in an image and the error in a curve obtained after a complex mathematical data reduction is not something that is easily achieved.*

Reviewer 3: In Figure 5 are shown the results for $m/z = 115$ (top), 70 (middle) and total (bottom row). Why are in the total data so many points missing? Here are basically the data from $m/z=70$ shown. And even the PES is an exact copy of the middle row (at least for left and middle column).

Authors answer: *The b_1 curves correspond to a normalized photoelectron angular distribution, ie to a normalization to the total cross section, obtained by dividing the raw dichroism data by the total (PES) intensity. Therefore, as the PES intensity vanishes, the b_1 values acquire huge error bars and are essentially meaningless (-> division by zero). This is why, for the sake of clarity, we have used an algorithm that suppresses the b_1 values plotting in the baseline regions outside the principal peak areas, with a 10 % threshold in general (15 % in some cases where the S/N is lower) with respect to the maximum PES intensity. [The value and need for such a procedure is already hinted on Fig.4c, for instance, when error bars clearly start to quickly increase to lower binding energies as the PES intensity starts to vanish.]*

We added in the caption of Fig. 5 : “PECD points corresponding to a relative PES intensity below 10 % (15% for the m/z 115 TD_{415} plot) of the maximum signal are not shown, due to their increasing error bars when normalized by the PES intensity”.

The reason for which we chose to apply a 15 % (instead of 10 %) cut-off threshold for the m/z 115 TD_{415} plot of panel (a) is obvious when looking at the figure below in which the plot has been done with a 10 % threshold. This clearly leads to the presence of 3 non-significant points affected by a huge error bar around 9.3 eV ionization energy. These points are meaningless and does bring any information since they correspond to vanishing cross sections (PES).

To be consistent, we also added in the caption of Fig. 4: “PECD points corresponding to a relative PES intensity below 10 % of the maximum signal are not shown, due to their increasing error bars when normalized by the PES intensity.”

The fact that the $m/z=70$ and the total data (PES and b_1 curve) are very similar is normal since by far the m/z 70 fragment dominates the mass spectrum at 10.2 eV, the parent (m/z 115) representing a very minor component of the mass spectrum (see Fig. S4), ie of the “total” signal.

Reviewer 3: Why aren't the experimental results from Figure 5 not overlaid with the theory from Figure 6?

Authors answer: There is here some misunderstanding: the data from fig 5 are the PECD ($2b_1$) and PES profiles as a function of binding energy at the single photon energy of 10.2 eV, ie corresponding to a fixed kinetic energy of ~ 1.45 eV for Type I conformer (HOMO orbital). Fig.6 shows theoretical modelling of b_1 for the various conformers as a function of varying electron kinetic energy. Then the

experiment/theory comparison is made in Fig.7 by considering at each photon energy (including 10.2 eV, ie the data from Fig.5), and for the 3 vaporization conditions the PES-weighted Gaussian fitted average b_1 values for the Type I conformer HOMO band (centred around 8.75 eV binding energy). See explanations in page 13&14. Therefore, to answer directly the reviewer, the data of Fig. 5 is overlaid with theory in Fig.7 (not Fig.6), as the second set of symbols at 1.45 eV KE.

Reviewer 3: In Figure 7, it seems that data points are missing. What happened to the red data point around 2 eV and why are the blue and green points not shown for higher energies, . . .

Authors answer: The colors represent three separate set of experiments, each of them performed on a different experimental run and with a different vaporization source. With the TD method, it is very difficult to go towards high KE and therefore high photon energy because of: (a) the presence of water (ionization Energy = 12.62 eV) in the beam, introduced as a solvent of the aerosols and which may not be completely removed; and (b) because of electrostatic difficulties in achieving a high enough electric field in the extraction region of the VMI spectrometer (to collect 100 % electrons) when the thermodesorber is present. This makes it challenging to collect data above 13.5 eV (4.75 eV kinetic energy for the type I HOMO band) which was only achieved for the TD₄₉₃. For the RH conditions, there is no thermodesorber, so it is possible to go towards high photon energies (although water from the chamber background can still be an issue, see Fig. S4), so we managed to collect data in the RH conditions up to 17.5 eV. It turns out that for the RH conditions we were more interested in the higher energies, so, because of beamtime limited access, we skipped the 10.8 eV measurement corresponding to 2.05 eV KE, to concentrate on higher photon energies. All of our measured energies for each vaporization conditions are summarized and visible in Fig. S4.

We thank the reviewer for bringing the “RH 10.8 eV case” to our attention and appreciate the opportunity to correct a plotting error in Figure 10. We now realized that in Fig. 10, there was a point for the RH condition at 10.8 eV although, as stated above, we did not perform any measurement at this energy. Looking at an earlier version of this figure, we found that during the optimization of its layout between several group members, the missing 10.8 eV RH entry in the data-table, which was initially flagged for the plotting software with a “NaN”, was simply cleared (left blank). As a consequence, when replotting this data-table the software no longer skipped the missing data cell as intended, but effectively left-shifted the rest of that row, plotting the 11.5 eV data at 10.8 eV, the 12.5 eV data at 11.5 eV etc..

We of course corrected, in the revised Figure 10 of the present version of the manuscript, this plotting error (not changing the actual survival rate dataset) which has no implication in any discussions or on the outcome of the paper.

Reviewer 3: while a fit to experimental data is shown. How can results be fitted, without data points?

Authors answer: In the text p14 we describe the fitting:

“To pursue this more quantitatively, we have taken an expression, $b_1(expt) = x \times b_1(C) + (1 - x) \times b_1(D)$, applying a least squares fit for x to each data set”.

We are thus performing a 1-parameter linear least squares fit of a model function (weighted combination of two pre-calculated curves) to a minimum of 5 experimental points for each sample condition. It may help to recognise that this is fully analogous to e.g. a least squares straight line fit to discrete data points, where the modelled best fit line can be used to both interpolate between experimental points, and to extrapolate beyond them. Of course, in any such case one’s confidence depends not just on the goodness of fit, but on the validity of the presumed model function.

Reviewer 3: Similarly in Figure 8. Where are the data points for green/blue at higher kinetic energies? They have obviously been recorded, as Figure 2 suggests. And in Figure 2 it would be nice to put some numbers on the y-axis - just to better show that the data are shown on a linear scale.

Authors answer: *The answer is again that no measurements were made above 13.5 eV photon energy with the TD (blue and green data), ie above 4 eV electron KE for Conformer Type II (allowing for its 9.65 eV binding energy).*

Figure 2 shows TPEPICO data which are obtained by scanning the photon energy and recording only threshold electrons (Zero KE.). Contrastingly, PECD data from Fig. 7 & 8 are obtained by recording all electron energies but only at a few fixed photon energies. Figures 2 and 8 thus show the results of very different measurement techniques and the suggested direct comparison is invalid. In particular the comment appears to be confusing the "Photon Energy" axis in Fig.2, and the "Electron Energy" axis in Fig.8. The ~4 eV upper limit (electron energy) of the blue/green data sets in Fig. 8 in fact corresponds to a photon energy of 13.5 eV once the 9.65 eV ionization energy is allowed for.

As for the linear scale of Fig.2, there the caption clearly notes

"The relative signals have been normalized according to the maximum of the TPES first band centred around 9.5 eV" (for the purposes of facilitating comparison).

Thus, the absolute intensity scale for the 3 spectra in each subplot is different. The axes are consequently presented in arbitrary units, but linearly spaced tics indicating a linear scale. This is very common practice.

Reviewer 3: Proline is an amino acid as the authors wrote. However, it naturally appears an inner salt / zwitterion. And within a 30 pages manuscript, I miss a discussion on this.

Authors answer: *The reviewer is right that in salt or in solution amino-acids may appear as zwitterions. However, isolated gas phase amino-acids such as proline are fully neutrals, with both neutral amino and carboxylic groups. This is the only form considered in the gas phase literature (see Ref. 9 for a short discussion).*

Therefore, in page 2 we added: "In dilute environments such as the interstellar medium, Pro, like other amino acids, is to be expected in its neutral form, unlike the zwitterionic forms found in condensed phases. Hence molecular structure of amino acids in the gas phase requires its own direct study."

Reviewer 3: It is essential to show that the molecule, which is investigated is safely brought intact into gas phase and not destroyed.

Authors answer:

We believe that the characterization of gas phase proline in the various vaporization conditions is already well discussed and documented in the Methods section and associated bibliography. Plus, we already address the thermal decomposition issue in response to reviewers 2 (point 2) in the former rebuttal letter:

"Regarding Fig S4 (former S1), an increase of temperature favors the fragmentation of the cation in a hot ground state (a very well documented effect of dissociative photoionization, see Ref. 71 and 72), not a thermal decomposition of the neutral (which we never observed with the TD method for any amino-acids). Would the latter process occur, the m/z 70 peak in the RH panels (c) would have a double structure: a sharp peak (for neutral decomposition) superimposed onto a broad one corresponding to the genuine dissociative ionization of intact neutral proline (KER effect). This is not the case. Also, by filtering the electron image with various part of this peak (center or wings) we find the same associated PECD, meaning that we are sampling the same neutral molecule: the intact parent Pro"

Reviewer 3: The mass specs in Figure 11 (a) and (b) clearly indicate that the hump towards higher masses that dimers/clusters are created, which then fragment -> Thus a larger mass, depending on the additional group is observed. And even the correct mass of a molecule, doesn't proof that the molecule is present in its shape/conformer, as expected. I miss a clear discussion on this point.

Authors answer:

The mass spectra of Fig.11 (a) and (b) corresponds to TD conditions. As mentioned in the caption of Fig.11, the broad and asymmetric shape of the peaks (both parent and fragments) are simply due to the very large interaction zone when using the TD associated with the inhomogeneous electric field used in the photoionization source for the electron VMI detection. This is not at all associated to the presence of near-monomer-mass fragments formed from unstable dimers/cluster. Actually, the energy releases that would be necessary to account for the observed peak broadening in such a scenario would be completely unfeasible (several eV). We performed checks that these broadened peaks do not in fact encompass varying masses/structures by comparing the electron spectra (PES and PECD) recorded in coincidence with ions sampled at different points across the peak widths.

The origin of the ToF peak broadening seen with the TD sample condition have been clarified by the addition of two sentences to the "Characterisation of vapour" section of the manuscript (p.23):

"The TD spectra show some asymmetric instrumental peak broadening due to ions from the spatially extended plume of desorbed sample experiencing a greater variability in the non-uniform source extraction field (required for the VMI electron detection). Checks were performed to establish that these broadened peaks are in fact homogeneous (do not encompass varying masses/structures) by examining the electron images (and hence PES and PECD spectra) recorded in coincidence with ions sampled at different points across the peak widths."

And then adding to the next sentence that the RH ToF spectrum "..., which is not so afflicted thanks to a tightly collimated molecular beam source."

Regarding the possibility of generating clusters with the TD method, we already answered this point in the first rebuttal letter (point 4 of reviewer 2), the answered is pasted below:

"It is true that the presence of clusters would affect the PES (and the PECD profile in fact). This is a well-documented effect (see ref 28 (now Ref. 34) for a collection of references on clustering effect on PECD). The usual effect on the PES is a red-shifting on the parent mass (due to cascading dissociation ionization processes) which we did not observe, nor in any trace of stable dimer at m/z 230. This is not surprising at all since even with the TD493 method (providing a plume of neutral not a molecular beam per se) the internal temperature was determined at 384 K (see Table 2), a temperature at which no Van der Waals nor H-bonded clusters can be formed and survive (this would require a quite strong adiabatic expansion leading to much colder temperatures to be formed, see for instance Nahon et al. Phys. Rev. A **82**, 032514 (2010) or Powis et al., PCCP **16**, 467 (2014))."

Reviewer 3: The discussion at the end of the manuscript regarding astrophysical relevance / homochirality is a completely detached topic from the paper. The obtained results are may be interesting for the future research here, but such a long discussion of things, which the authors didn't work on, is only nice to read, but not of any relevance.

Authors answer: As we already addressed this issue in response to Reviewer 3 in the first rebuttal letter, we consider this discussion on astrophysical relevance in very tight connection, in fact as a direct outcome, with the rest of the paper since it involves all aspects of our study: the conformer-dependent fragmentation pattern, the conformer-dependent PECD, the internal temperature modeling, and the PECD theoretical modelling for temperature effects extrapolation.

In addition, we believe that the broad readership of Commun. Chem. will find this fundamental discussion not only nice to read but also relevant with clear connections with astrochemistry, biochemistry and the origin of life, as it was for instance the case of reviewer 1 who showed a genuine interest in this discussion. We note that the origin of life is a topic covered by Commun. Chem. (see

the recent paper: Butchet al. Open questions in understanding life's origins. *Commun Chem* 4, 11 (2021)).

However, in order to make this part more clearly identified as an intrinsic motivation at the core of the present study, we modified the introduction by adapting and moving earlier (now as a 3rd paragraph) in page 2, the introductory paragraph on astrochemistry, which now reads:

"A motivation for the current work has been the continuation and extension of our previous studies on the VUV photoionization of alanine enantiomers^{5,6} which adduced a potential new scenario for postulated astrophysical origins of life's homochirality.⁷ Both alanine (Ala) and proline (Pro) belong to the first five amino acids to have been recruited into the genetic code,⁸ and have also been detected in the Murchison meteorite in large quantities and with an excess of the L enantiomer.⁹ Hence it will be of considerable interest to seek to establish whether similar properties apply for the chiral VUV photoionization of Pro, as was tentatively suggested.¹⁰ In dilute environments such as the interstellar medium, Pro, like other amino acids, is to be expected in its neutral form, unlike the zwitterionic forms found in condensed phases. Hence molecular structure of amino acids in the gas phase requires its own direct study."

We also slightly changed the last sentence of the introduction which reads now as:

"Finally, the astrophysical PECD-based scenario for the origin of life's homochirality is addressed, with this conformer-dependent analysis on Pro used to examine possible temperature constraints and implications relevant to interstellar medium conditions."

Note that we took advantage of this second round of review to introduce in the astrochemical implication section, in page 18, a new reference on serine (Hartweg et al, *J. Phys Chem. Letters*, submitted)], as: "...and most of all of the same sign (negative for the L-enantiomer) as Ala¹⁰ and most likely as Ser.⁷¹"

2- On more general issues:

Reviewer 3: The Paper has interesting results to be present, which should be published. However, I still think that these are only of interest for a small group of specialists and not a broader audience.

Authors answer: We don't share this opinion, and we already strongly defended this position in the first rebuttal letter addressing the same concerns by the same reviewer.

Reviewer 3: I still think that the manuscript doesn't have any significant contribution to the field. Neither PECD is a new tool, nor the molecule is new, nor is the gained knowledge new.

It is known since a long time that proline has 4 conformers and it is obvious that each of the conformers shows a different PECD, when correctly prepared. And this is the only contribution of this study. The experiment and the results are rather straight forward in this community. Conformer dependent PECD has already been measured 10 years ago (reference 30 and 32), as the authors already stated - on alaninol, which is an amino alcohol, instead of proline (an amino acid). Also the separation of conformers depending on the ionization has been shown in the past.

Authors answer: Novelty and interest of Proline were already raised by the same reviewer in the first round, and we already answered these points in the first rebuttal letter. The vast bulk of the chemical literature must surely address already known molecules. What is perhaps more pertinent in determining "interest" is the wider significance of a molecule. On this score proline is not readily dismissed. It is a major proteic amino-acid (as already argued) with very specific structural properties in link to its conformational landscape (see previous rebuttal letter) and this is the first VUV photodynamics and PECD study on this crucial system. In solution, proline adopts a very different structural form (zwitterionic) compared to the expected isolated gas-phase (neutral amino acid). Thus, any prior conformer knowledge or expectations obtained in solution phase cannot be automatically assumed to transfer to the gas phase. Transferring proline to the gas phase in the laboratory, as we

have done, is not trivial because it is thermolabile. Also, as our results (and literature comparisons) show, the reliable computational prediction of conformer stabilities is at the very limit (or even beyond) currently feasible electronic structure calculations. Consequently, the development and demonstration of alternative experimental/ computational spectroscopies, (such as PECD which we demonstrate here) is of clear value and interest.

PECD is not a new effect, but it is still undergoing development along different avenues with one aim, in particular, to become an analytical method in bio-chemistry (far beyond the case of proline). One of the most spectacular PECD properties is its very high sensitivity to conformations. But previous PECD studies (including indeed Ref. 30 (now Ref. 35) on alaninol, but also Ref. 31 (now Ref. 36) and 34 (now Ref. 6), as well as: G. Garcia, et al., *J. Phys. Chem. A* **114**, 847 (2010) and S. Daly et al., *Angew. Chem. Int. Ed. Engl.* **55**, 11054 (2016)) have been restricted to examining an ensemble of conformers, often at ill-defined temperatures and indeed not necessarily even at thermal equilibrium. Under these circumstances the conformer ratios were generally uncertain (a statistical Boltzmann-average ensemble of conformers was not guaranteed), limiting the scope for interpretation. The only way to develop investigation of possible conformational effect in these studies, in a PECD context, was to attempt to play with temperature. In the present study, by using the kinetic energy electron tagging, we can directly separate Type I and Type II conformers, and benchmark theoretical calculations for the two types of conformers. And that's new. Moreover, in the future, laser multiphoton PECD experiment will likely offer increased opportunities to achieve conformer-specific measurements, so that an understanding of the scope of PECD for structure interpretation will acquire more relevance.

There is nothing "obvious" in the fact that the 4 conformers should show different PECD. But it is one outcome of the paper that finds, for instance, a spectacular opposite sign PECD measured between type I and Type II conformers at 10.2 eV.

It is indeed true that a starting point for this study was the inferred separation of gas phase proline conformers by ionization (Ref. 9, now Ref. 15), but this lacked the specific experimental corroboration that PECD here provides, and (as already mentioned above) currently feasible electronic structure calculations fail to provide a consistent picture of the conformer energetics. Hence there is a clear need for experimental conformer identification methods to be developed.

We don't understand what means "correctly prepared". There is maybe some misunderstanding here. We don't prepare the conformers, they come naturally with a thermal distribution, we only vary the temperature which in turn varies the conformer population

Besides, these i2PEPICO experiments on a genuine amino-acid, known to be very difficult to bring in the gas phase, are not all straightforward but constitute the state-of-the-art for the community.

REVIEWERS' COMMENTS:

Reviewer #3 (Remarks to the Author):

see attached file

We are pleased that the first two reviewers are satisfied with the last version of the manuscript appearing to them as suitable for publication in *Commun. Chem.* Reviewer 3 is, however, still raising concerns, several of which are simply a repetition of issues that have previously been raised. While it is of course open to a reviewer to come back on points where a previous response was not considered satisfactory, we find it unhelpful that in repeating questions here there is no recognition or acknowledgement of our previous responses. We do not then know whether these were ignored or were considered deficient in some way. If the latter we are denied the opportunity to attempt further manuscript changes which might help improve the accessibility of our paper to a wider, non-specialist readership (something we would be happy to do). We do, however, take the opportunity to address and avert some misconceptions and misunderstandings of the experimental methodology that are apparent to us in some of the comments. Below are our answers (in *italic*) and revisions (underlined in yellow) to each of Reviewer 3's comments:

I thank the authors for their extended discussions and corrections. Please see my comments at each point they answered, highlighted in green (ok) or yellow (question, problem, to do)

Reviewer #3 (Remarks to the Author):

1-On the technical issues

Reviewer 3: It is surprising that most of the figures seem to show different amounts of data points, which is questionable. Did the authors removed certain data points, as they didn't fit? Let me give some examples:

Authors answer: We note the implication of inappropriate data manipulation and we are happy to be able hereafter to refute and fully address all these concerns.

Reviewer 3: Why are in Figure 4 (b) and (c) not the inverted graphs for the other enantiomer shown?

Authors answer: The same reviewer asked the same question in the first evaluation round regarding the then Fig.6 (now Fig. 5). As we responded in the first review: "In an ideal world, having dual L and D data for each photon energy would be great. In practice, these experiments are time consuming and synchrotron beamtime is not infinite. We chose to acquire a set of data for each enantiomer but at different photon energies, to cover a wide VUV range, with a single control photon energy at 8.7 eV (see Fig.4(a)) in order to check and show the nice L/D mirroring of the data. This is a common practice in most of the PECD literature" (see for instance Nahon et al. *PCCP* **18**, 12696 (2016)). However, for the sake of clarity we now add in the Methods section, page 23 & 24: - A new subheading "PECD measurements procedure" - A new sentence, at the end of this subsection: "We used a single control-energy (8.7 eV) for checking the satisfactory chiral mirroring by repeating measurements with the L- and D-enantiomers; at all other photon energies, PECD was investigated using a single enantiomer, the selection of which was alternated between steps in photon energy in order to cover a greater total range and so optimize the beamtime use."

OK, fine with me

Reviewer 3: Also the comment "very satisfactory in the description of Figure 4 (a) stress this point a bit to much. Both curves here show a similar trend, decreasing towards 0, but it seems that within the error bars a mirroring of the curves would show significant discrepancies, as the blue for example shows an oscillation, which is not present in the red curve..."

Authors answer: We agree with the reviewer that the mirroring in Fig. 4(a) is slightly outside of the error bars. This is because the error bars take into account only statistical errors, as explained at the bottom of page 23 in the Methods section. There are probably some additional limited systematical errors, such as imperfect and different enantiopurity of the samples, or different S/N for both enantiomers due to the preparation of different solutions to be nebulized, which may lead to a slightly non-zero baseline between the two enantiomers as often encountered in CD experiments. We could have normalized the data of Fig. 4a, as often done in absorption CD works, so that the mirroring would be better, but we did not, nor did we smooth the data. In order to take into account the remark of the reviewer, in the caption of Fig.4, we changed "the very satisfactory expected L/D mirroring" into "the satisfactory and expected L/D mirroring".

OK, fine with me

Reviewer 3: ...The error bars seems not being real. Why? Rule of thumb is that for a statistical error on average every 3rd point should be displaced out of the error bars, while here the curves are surprisingly smooth.

Authors answer: Our derivation of error bars by standard statistical treatments is properly described in the Methods section (p. 23): “The statistical error bars on the dichroic parameter b_1 are given as the standard error on the principle that each image pixel acts as an independent counter that follows a Poisson distribution, with the associated error properly propagated through all subsequent operations.” This is a widely recognized, statistically based approach to data treatment, whose principles and formulae are fully documented in many textbooks. An arbitrary variation of the error bars to satisfy some undocumented rule of thumb would not be appropriate. We do, however, clearly state at points in the manuscript that these statistical error bars exclude any systematic error.

The authors just wrote what they did and unfortunately do not explain why the curve is much smoother than it should be. I would have expected a more critical reply than “We have done it this way and we don’t believe the rule of thumb”. This rule of thumb is nothing undocumented, it follows directly from the definition of standard deviation from the average value (which is what you see with your eyes as a smooth line): For normally distributed random numbers 68 % are within the $x \pm \sigma$, which means that 32 % are out of this interval. This means that roughly every 3rd point is out of the interval of 1σ .

The results might be right and the calculation of the errorbars might be correct as well, but it is more than surprising. However, I agree that the statistical error is most likely not the main source of error in all these measurements, as the authors stated in their reply above. And I miss here (as at most other positions) that the authors are clear about this point. Coming back to Fig. 4, if the systematic errors are larger than the statistical ones, they should clearly be shown in the figure or at least in the figure description. I haven’t read the whole manuscript from beginning sentence by sentence, but searching I found only once the word “systematic” as in systematic error (page 15) and only in Fig. 10 “statistical error”, otherwise it is always stated as “error”.

Reviewer 3: Also the insets in (b) and (c) seem more noisy than the shown PECD is. Or are the insets just a subset?

Authors answer: The insets in Fig.4 (b) and (c) are raw difference images from which, after Abel inversion, the b_1 curve is obtained. The electron counts in the ~65000 pixels in an image are effectively combined and reduced by the data treatment to a much smaller small number of $b_1(E)$ parameters. The statistical (Poisson) noise seen in individual pixels is thus quantitatively transformed in the data reduction to the uncertainty of those smaller number of parameters, as described in the previous response. Hence, an intuitive visual comparison of the “pixel noise” in an image and the error in a curve obtained after a complex mathematical data reduction is not something that is easily achieved.

OK, fine with me

Reviewer 3: In Figure 5 are shown the results for $m/z = 115$ (top), 70 (middle) and total (bottom row). Why are in the total data so many points missing? Here are basically the data from $m/z=70$ shown. And even the PES is an exact copy of the middle row (at least for left and middle column).

Authors answer: The b_1 curves correspond to a normalized photoelectron angular distribution, ie to a normalization to the total cross section, obtained by dividing the raw dichroism data by the total (PES) intensity. Therefore, as the PES intensity vanishes, the b_1 values acquire huge error bars and are essentially meaningless (-> division by zero). This is why, for the sake of clarity, we have used an algorithm that suppresses the b_1 values plotting in the baseline regions outside the principal peak areas, with a 10 % threshold in general (15 % in some cases where the S/N is lower) with respect to the maximum PES intensity. [The value and need for such a procedure is already hinted on Fig.4c, for instance, when error bars clearly start to quickly increase to lower binding energies as the PES intensity starts to vanish.] We added in the caption of Fig. 5 : “PECD points corresponding to a relative PES intensity below 10 % (15% for the m/z 115 TD₄₁₅ plot) of the maximum signal are not shown, due to their increasing error bars when normalized by the PES intensity”.

The reason for which we chose to apply a 15 % (instead of 10 %) cut-off threshold for the m/z 115 TD₄₁₅ plot of panel (a) is obvious when looking at the figure below in which the plot has been done with a 10 % threshold. This

clearly leads to the presence of 3 non-significant points affected by a huge error bar around 9.3 eV ionization energy. These points are meaningless and does bring any information since they correspond to vanishing cross sections (PES). To be consistent, we also added in the caption of Fig. 4: “PECD points corresponding to a relative PES intensity below 10 % of the maximum signal are not shown, due to their increasing error bars when normalized by the PES intensity.” The fact that the $m/z=70$ and the total data (PES and b_1 curve) are very similar is normal since by far the m/z 70 fragment dominates the mass spectrum at 10.2 eV, the parent (m/z 115) representing a very minor component of the mass spectrum (see Fig. S4), ie of the “total” signal.

I agree in general. However the authors could also combine 3 data points to one (as they would have done anyways, if the photon energy resolution would be not as good as they had). Therefore the intensity per point is about 3 times larger, the error bars $\sqrt{3}$ smaller and the data could be presented. Obviously leaving out points gives the readers a bitter aftertaste.

Reviewer 3: Why aren't the experimental results from Figure 5 not overlaid with the theory from Figure 6?

Authors answer: There is here some misunderstanding: the data from fig 5 are the PECD ($2b_1$) and PES profiles as a function of binding energy at the single photon energy of 10.2 eV, ie corresponding to a fixed kinetic energy of ~ 1.45 eV for Type I conformer (HOMO orbital). Fig.6 shows theoretical modelling of b_1 for the various conformers as a function of varying electron kinetic energy. Then the experiment/theory comparison is made in Fig.7 by considering at each photon energy (including 10.2 eV, ie the data from Fig.5), and for the 3 vaporization conditions the PES-weighted Gaussian fitted average b_1 values for the Type I conformer HOMO band (centred around 8.75 eV binding energy). See explanations in page 13&14. Therefore, to answer directly the reviewer, the data of Fig. 5 is overlaid with theory in Fig.7 (not Fig.6), as the second set of symbols at 1.45 eV KE.

OK, my fault. Thanks for solving the misunderstanding.

Reviewer 3: In Figure 7, it seems that data points are missing. What happened to the red data point around 2 eV and why are the blue and green points not shown for higher energies, ...

Authors answer: The colors represent three separate set of experiments, each of them performed on a different experimental run and with a different vaporization source. With the TD method, it is very difficult to go towards high KE and therefore high photon energy because of: (a) the presence of water (Ionization Energy = 12.62 eV) in the beam, introduced as a solvent of the aerosols and which may not be completely removed; and (b) because of electrostatic difficulties in achieving a high enough electric field in the extraction region of the VMI spectrometer (to collect 100 % electrons) when the thermodesorber is present. This makes it challenging to collect data above 13.5 eV (4.75 eV kinetic energy for the type I HOMO band) which was only achieved for the TD₄₉₃. For the RH conditions, there is no thermodesorber, so it is possible to go towards high photon energies (although water from the chamber background can still be an issue, see Fig. S4), so we managed to collect data in the RH conditions up to 17.5 eV. It turns out that for the RH conditions we were more interested in the higher energies, so, because of beamtime limited access, we skipped the 10.8 eV measurement corresponding to 2.05 eV KE, to concentrate on higher photon energies. All of our measured energies for each vaporization conditions are summarized and visible in Fig. S4. We thank the reviewer for bringing the “RH 10.8 eV case” to our attention and appreciate the opportunity to correct a plotting error in Figure 10. We now realized that in Fig. 10, there was a point for the RH condition at 10.8 eV although, as stated above, we did not perform any measurement at this energy. Looking at an earlier version of this figure, we found that during the optimization of its layout between several group members, the missing 10.8 eV RH entry in the data-table, which was initially flagged for the plotting software with a “NaN”, was simply cleared (left blank). As a consequence, when replotting this data-table the software no longer skipped the missing data cell as intended, but effectively left-shifted the rest of that row, plotting the 11.5 eV data at 10.8 eV, the 12.5 eV data at 11.5 eV etc.. We of course corrected, in the revised Figure 10 of the present version of the manuscript, this plotting error (not changing the actual survival rate dataset) which has no implication in any discussions or on the outcome of the paper.

OK, thanks for the explanation and the corrections for Fig. 10, which raised some of my concerns.

Reviewer 3: while a fit to experimental data is shown. How can results be fitted, without data points?

Authors answer: In the text p14 we describe the fitting: "To pursue this more quantitatively, we have taken an expression, $b^{\text{expt}} = x \times b^{\text{C}} + 1 - x \times b^{\text{D}}$, applying a least squares fit for x to each data set". We are thus performing a 1-parameter linear least squares fit of a model function (weighted combination of two pre-calculated curves) to a minimum of 5 experimental points for each sample condition. It may help to recognise that this is fully analogous to e.g. a least squares straight line fit to discrete data points, where the modelled best fit line can be used to both interpolate between experimental points, and to extrapolate beyond them. Of course, in any such case one's confidence depends not just on the goodness of fit, but on the validity of the presumed model function.

In general fine with me. However the color coding and description of the broken lines suggests that the lines have more in common with the data points than they actually have. The broken lines are basically an average of the theory curves folded with a value extracted from the data points. The lines are neither pure theory nor pure experiment. In order to avoid misinterpretation here and being fair presenting results, I ask the authors putting the explanation of the broken lines separated from the points in the inset box.

Reviewer 3: Similarly in Figure 8. Where are the data points for green/blue at higher kinetic energies? They have obviously been recorded, as Figure 2 suggests. And in Figure 2 it would be nice to put some numbers on the y-axis - just to better show that the data are shown on a linear scale.

Authors answer: The answer is again that no measurements were made above 13.5 eV photon energy with the TD (blue and green data), ie above 4 eV electron KE for Conformer Type II (allowing for its 9.65 eV binding energy). Figure 2 shows TPEPICO data which are obtained by scanning the photon energy and recording only threshold electrons (Zero KE.). Contrastingly, PECD data from Fig. 7 & 8 are obtained by recording all electron energies but only at a few fixed photon energies. Figures 2 and 8 thus show the results of very different measurement techniques and the suggested direct comparison is invalid. In particular the comment appears to be confusing the "Photon Energy" axis in Fig.2, and the "Electron Energy" axis in Fig.8. The ~4 eV upper limit (electron energy) of the blue/green data sets in Fig. 8 in fact corresponds to a photon energy of 13.5 eV once the 9.65 eV ionization energy is allowed for. As for the linear scale of Fig.2, there the caption clearly notes "The relative signals have been normalized according to the maximum of the TPES first band centred around 9.5 eV" (for the purposes of facilitating comparison). Thus, the absolute intensity scale for the 3 spectra in each subplot is different. The axes are consequently presented in arbitrary units, but linearly spaced ticks indicating a linear scale. This is very common practice.

I agree with the authors that, as done in Fig. 2, it is common practice to present in arbitrary units. However it is also common practice to have written a number on this axis (as "arbitrary units" doesn't mean anything); just put 1, 2, 3, 4 on the y-axis and its fine. Also I do not understand than why in Figure 7 the 3rd data point is "missing" (where blue/green are shown), while in figure 8 the 2nd data point is missing? For this, please see figure below.

Another point. The distinction between the conformers type I and II is done by the electron's kinetic energy. This is basically an offset of 0.5 – 0.7 eV as can be seen from table 1. This is also roughly the difference between the right most point in fig. 7 and 8. But why is this not a constant offset, rather than an increasing offset, when comparing 7 and 8 (see figure below)?

Reviewer 3: Proline is an amino acid as the authors wrote. However, it naturally appears an inner salt / zwitterion. And within a 30 pages manuscript, I miss a discussion on this.

Authors answer: The reviewer is right that in salt or in solution amino-acids may appear as zwitterions. However, isolated gas phase amino-acids such as proline are fully neutrals, with both neutral amino and carboxylic groups. This is the only form considered in the gas phase literature (see Ref. 9 for a short discussion).

Therefore, in page 2 we added: “In dilute environments such as the interstellar medium, Pro, like other amino acids, is to be expected in its neutral form, unlike the zwitterionic forms found in condensed phases. Hence molecular structure of amino acids in the gas phase requires its own direct study.”

OK, thanks for clarifying this point.

Reviewer 3: It is essential to show that the molecule, which is investigated is safely brought intact into gas phase and not destroyed.

Authors answer: We believe that the characterization of gas phase proline in the various vaporization conditions is already well discussed and documented in the Methods section and associated bibliography. Plus, we already address the thermal decomposition issue in response to reviewers 2 (point 2) in the former rebuttal letter: “Regarding Fig S4 (former S1), an increase of temperature favors the fragmentation of the cation in a hot ground state (a very well documented effect of dissociative photoionization, see Ref. 71 and 72), not a thermal decomposition of the neutral (which we never observed with the TD method for any amino-acids). Would the latter process occur, the m/z 70 peak in the RH panels (c) would have a double structure: a sharp peak (for neutral decomposition) superimposed onto a broad one corresponding to the genuine dissociative ionization of intact neutral proline (KER effect). This is not the case. Also, by filtering the electron image with various part of this peak (center or wings) we find the same associated PECD, meaning that we are sampling the same neutral molecule: the intact parent Pro”

OK, I missed this point from reviewer 2. And I agree that for RH the molecule stays at least intact, which is obviously not the case for TD, where the peak is much broader and additional shows higher masses (from clusters/dimers, which asymmetrically dissociated. From H_2O dimers/clusters it is well known that in a mass spectrum one dominantly observes HO^+ and $(H_2O)_nH^+$). The authors haven't ruled out problems with clusters.

Reviewer 3: The mass specs in Figure 11 (a) and (b) clearly indicate that the hump towards higher masses that dimers/clusters are created, which then fragment -> Thus a larger mass, depending on the additional group is observed. And even the correct mass of a molecule, doesn't proof that the molecule is present in its shape/conformer, as expected. I miss a clear discussion on this point.

Authors answer: The mass spectra of Fig.11 (a) and (b) corresponds to TD conditions. As mentioned in the caption of Fig.11, the broad and asymmetric shape of the peaks (both parent and fragments) are simply due to the very large interaction zone when using the TD associated with the inhomogeneous electric field used in the photoionization source for the electron VMI detection. This is not at all associated to the presence of near-monomer-mass fragments formed from unstable dimers/cluster. Actually, the energy releases that would be necessary to account for the observed peak broadening in such a scenario would be completely unfeasible (several eV). We performed checks that these broadened peaks do not in fact encompass varying masses/structures by comparing the electron spectra (PES and PECD) recorded in coincidence with ions sampled at different points across the peak widths. The origin of the ToF peak broadening seen with the TD sample condition have been clarified by the addition of two sentences to the “Characterisation of vapour” section of the manuscript (p.23): “The TD spectra show some asymmetric instrumental peak broadening due to ions from the spatially extended plume of desorbed sample experiencing a greater variability in the non-uniform source extraction field (required for the VMI electron detection). Checks were performed to establish that these broadened peaks are in fact homogeneous (do not encompass varying masses/structures) by examining the electron images (and hence PES and PECD spectra) recorded in coincidence with ions sampled at different points across the peak widths.” And then adding to the next sentence that the RH ToF spectrum “..., which is not so afflicted thanks to a tightly collimated molecular beam source.” Regarding the possibility of generating clusters with the TD method, we already answered this point in the first rebuttal letter (point 4 of reviewer 2), the answered is pasted below: “ It is true that the presence of clusters would affect the PES (and the PECD profile in fact). This is a well-documented effect (see ref 28

(now Ref. 34) for a collection of references on clustering effect on PECD). The usual effect on the PES is a red-shifting on the parent mass (due to cascading dissociation ionization processes) which we did not observe, nor in any trace of stable dimer at m/z 230. This is not surprising at all since even with the TD493 method (providing a plume of neutral not a molecular beam per se) the internal temperature was determined at 384 K (see Table 2), a temperature at which no Van der Waals nor H-bonded clusters can be formed and survive (this would require a quite strong adiabatic expansion leading to much colder temperatures to be formed, see for instance Nahon et al. Phys. Rev. A **82**, 032514 (2010) or Powis et al., PCCP **16**, 467 (2014)).”

I don't know the setup exactly and therefore cannot check that “large interaction zone when using the TD associated with the inhomogeneous electric field used” is the source. A check with Argon/Krypton would help here. However, it looks strange that the peak is asymmetric with a shape that rather looks like two peaks, with one being factor 2 larger than the second one, which is shifted by a few amu. The water peak in Figure S4 TD493 at 13.5 eV has much less of a low plateau on the right side. And this little plateau is also present in the RH data, which suggests that not the TD-technique is solely the source of the broadening. Also I agree that the broadening of the peak does not originate from the kinetic energy of a dissociating (parent/fragment) ion. There is not enough energy, as the authors also said, and it would make the peak symmetrically broader. In Tia et al. J. Phys. Chem. Lett. **4**, 2698-2704 (2013) Fig. 1b there is for alanine with the TD technique also a broader peak shown, but there the amount of “larger” masses is much less than in the present paper, or in J. Phys. Chem. Lett. **12**, 2385-2393 (2021) Fig. 1 where Serine molecules are TD and no asymmetric peak broadening at all is visible. This suggests that there more than purely a problem of the imaging.

Reviewer 3: The discussion at the end of the manuscript regarding astrophysical relevance / homochirality is a completely detached topic from the paper. The obtained results are may be interesting for the future research here, but such a long discussion of things, which the authors didn't work on, is only nice to read, but not of any relevance.

Authors answer: As we already addressed this issue in response to Reviewer 3 in the first rebuttal letter, we consider this discussion on astrophysical relevance in very tight connection, in fact as a direct outcome, with the rest of the paper since it involves all aspects of our study: the conformer-dependent fragmentation pattern, the conformer-dependent PECD, the internal temperature modeling, and the PECD theoretical modelling for temperature effects extrapolation. In addition, we believe that the broad readership of Commun. Chem. will find this fundamental discussion not only nice to read but also relevant with clear connections with astrochemistry, biochemistry and the origin of life, as it was for instance the case of reviewer 1 who showed a genuine interest in this discussion. We note that the origin of life is a topic covered by Commun. Chem. (see the recent paper: Butchet al. Open questions in understanding life's origins. Commun Chem **4**, 11 (2021)). However, in order to make this part more clearly identified as an intrinsic motivation at the core of the present study, we modified the introduction by adapting and moving earlier (now as a 3rd paragraph) in page 2, the introductory paragraph on astrochemistry, which now reads: “A motivation for the current work has been the continuation and extension of our previous studies on the VUV photoionization of alanine enantiomers^{5,6} which adduced a potential new scenario for postulated astrophysical origins of life's homochirality.⁷ Both alanine (Ala) and proline (Pro) belong to the first five amino acids to have been recruited into the genetic code,⁸ and have also been detected in the Murchison meteorite in large quantities and with an excess of the L enantiomer.⁹ Hence it will be of considerable interest to seek to establish whether similar properties apply for the chiral VUV photoionization of Pro, as was tentatively suggested.¹⁰ In dilute environments such as the interstellar medium, Pro, like other amino acids, is to be expected in its neutral form, unlike the zwitterionic forms found in condensed phases. Hence molecular structure of amino acids in the gas phase requires its own direct study.” We also slightly changed the last sentence of the introduction which reads now as: “Finally, the astrophysical PECD-based scenario for the origin of life's homochirality is addressed, with this conformer-dependent analysis on Pro used to examine possible temperature constraints and implications relevant to interstellar medium conditions.” Note that we took advantage of this second round of review to introduce in the astrochemical implication section, in page 18, a new reference on serine (Hartweg et al, J. Phys Chem. Letters, submitted)], as: “...and most of all of the same sign (negative for the L-enantiomer) as Ala¹⁰ and most likely as Ser.⁷¹”

I value the changes in the introduction and I appreciate the results, presented in Fig. 9 and 10. But still, I don't see, why they are discussed in a context of astrochemistry (as in so many other publications from the authors already); it

seems to me just making the PECD results bigger than they are by adding a long discussion on unrelated things. That a photoionized molecule dissociates and that this molecule dissociates more when more energy is absorbed is just way to obvious (Fig. 10). I'm sorry, but here we will not find an agreement.

On more general issues:

Reviewer 3: The Paper has interesting results to be present, which should be published. However, I still think that these are only of interest for a small group of specialists and not a broader audience.

Authors answer: We don't share this opinion, and we already strongly defended this position in the first rebuttal letter addressing the same concerns by the same reviewer.

Sorry, but I do not agree and stick to my personal opinion as a scientists in this field.

Reviewer 3: I still think that the manuscript doesn't have any significant contribution to the field. Neither PECD is a new tool, nor the molecule is new, nor is the gained knowledge new. It is known since a long time that proline has 4 conformers and it is obvious that each of the conformers shows a different PECD, when correctly prepared. And this is the only contribution of this study. The experiment and the results are rather straight forward in this community. Conformer dependent PECD has already been measured 10 years ago (reference 30 and 32), as the authors already stated - on alaninol, which is an amino alcohol, instead of proline (an amino acid). Also the separation of conformers depending on the ionization has been shown in the past.

Authors answer: Novelty and interest of Proline were already raised by the same reviewer in the first round, and we already answered these points in the first rebuttal letter. The vast bulk of the chemical literature must surely address already known molecules. What is perhaps more pertinent in determining "interest" is the wider significance of a molecule. On this score proline is not readily dismissed. It is a major proteic amino-acid (as already argued) with very specific structural properties in link to its conformational landscape (see previous rebuttal letter) and this is the first VUV photodynamics and PECD study on this crucial system. In solution, proline adopts a very different structural form (zwitterionic) compared to the expected isolated gas-phase (neutral amino acid). Thus, any prior conformer knowledge or expectations obtained in solution phase cannot be automatically assumed to transfer to the gas phase. Transferring proline to the gas phase in the laboratory, as we have done, is not trivial because it is thermolabile. Also, as our results (and literature comparisons) show, the reliable computational prediction of conformer stabilities is at the very limit (or even beyond) currently feasible electronic structure calculations. Consequently, the development and demonstration of alternative experimental/ computational spectroscopies, (such as PECD which we demonstrate here) is of clear value and interest. PECD is not a new effect, but it is still undergoing development along different avenues with one aim, in particular, to become an analytical method in biochemistry (far beyond the case of proline). One of the most spectacular PECD properties is its very high sensitivity to conformations. But previous PECD studies (including indeed Ref. 30 (now Ref. 35) on alaninol, but also Ref. 31 (now Ref. 36) and 34 (now Ref. 6), as well as: G. Garcia, et al., *J. Phys. Chem. A* **114**, 847 (2010) and S. Daly et al., *Angew. Chem. Int. Ed. Engl.* **55**, 11054 (2016)) have been restricted to examining an ensemble of conformers, often at ill-defined temperatures and indeed not necessarily even at thermal equilibrium. Under these circumstances the conformer ratios were generally uncertain (a statistical Boltzmann average ensemble of conformers was not guaranteed), limiting the scope for interpretation. The only way to develop investigation of possible conformational effect in these studies, in a PECD context, was to attempt to play with temperature. In the present study, by using the kinetic energy electron tagging, we can directly separate Type I and Type II conformers, and benchmark theoretical calculations for the two types of conformers. And that's new. Moreover, in the future, laser multiphoton PECD experiment will likely offer increased opportunities to achieve conformer-specific measurements, so that an understanding of the scope of PECD for structure interpretation will acquire more relevance. There is nothing "obvious" in the fact that the 4 conformers should show different PECD. But it is one outcome of the paper that finds, for instance, a spectacular opposite sign PECD measured between type I and Type II conformers at 10.2 eV. It is indeed true that a starting point for this study was the inferred separation of gas phase proline conformers by ionization (Ref. 9, now Ref. 15), but this lacked the specific experimental corroboration that PECD here provides, and (as already mentioned above) currently feasible electronic structure calculations fail to provide a consistent picture of the conformer energetics. Hence there is a clear need for experimental conformer identification methods to be developed. We don't understand what means "correctly prepared". There is maybe some misunderstanding here.

We don't prepare the conformers, they come naturally with a thermal distribution, we only vary the temperature which in turn varies the conformer population Besides, these i2PEPICO experiments on a genuine amino-acid, known to be very difficult to bring in the gas phase, are not all straightforward but constitute the state-of-the-art for the community.

As I said, I don't question that the results should be published. But still, I don't see their novelty, necessary to be published in this journal. Here is nothing surprising or new, except that theory and experiment disagree quite a lot to my naive view and this is not surprising. I agree that the molecule has not that much been studied and has some interesting features, but still I don't see what one really learns from these results?

- The shape of the b1 parameter? -> New and novel yes, but nothing fundamental to learn from.

- Theory vs. experiment? -> Do not agree; theory seems still being many years behind.

- That the conformers of the molecule have been measured? Yes, new for this molecule, but not in general.

- That the molecule fragments after irradiation and fragments more when irradiated with higher energy? Not surprising at all.

Reviewer 3 (former comment): ...The error bars seems not being real. Why? Rule of thumb is that for a statistical error on average every 3rd point should be displaced out of the error bars, while here the curves are surprisingly smooth.

Former Authors answer:

Our derivation of error bars by standard statistical treatments is properly described in the Methods section (p. 23): “The statistical error bars on the dichroic parameter b_1 are given as the standard error on the principle that each image pixel acts as an independent counter that follows a Poisson distribution, with the associated error properly propagated through all subsequent operations.” This is a widely recognized, statistically based approach to data treatment, whose principles and formulae are fully documented in many textbooks. An arbitrary variation of the error bars to satisfy some undocumented rule of thumb would not be appropriate. We do, however, clearly state at points in the manuscript that these statistical error bars exclude any systematic error.

Reviewer 3 (new comment): The authors just wrote what they did and unfortunately do not explain why the curve is much smoother than it should be. I would have expected a more critical reply than “We have done it this way and we don’t believe the rule of thumb”. This rule of thumb is nothing undocumented, it follows directly from the definition of standard deviation from the average value (which is what you see with your eyes as a smooth line): For normally distributed random numbers 68 % are within the $x \pm \sigma$, which means that 32 % are out of this interval. This means that roughly every 3rd point is out of the interval of 1σ .

The results might be right and the calculation of the error bars might be correct as well, but it is more than surprising. However, I agree that the statistical error is most likely not the main source of error in all these measurements, as the authors stated in their reply above. And I miss here (as at most other positions) that the authors are clear about this point. Coming back to Fig. 4, if the systematic errors are larger than the statistical ones, they should clearly be shown in the figure or at least in the figure description. I haven’t read the whole manuscript from beginning sentence by sentence, but searching I found only once the word “systematic” as in systematic error (page 15) and only in Fig. 10 “statistical error”, otherwise it is always stated as “error”.

New Authors answer: Our responses stated very clearly that our error estimates are rooted in the random counting statistics of electrons in each pixel. These have, as we stated, a Poisson distribution. The Poisson distribution is not a Normal distribution as the referee now supposes in offering justification of his/her "rule of thumb". For example, at low mean the Poisson distribution is strongly skewed and asymmetric, and the variance differs from the Normal distribution.

Anyway, in order to be more precise, we added in the caption of Fig. 4, 8 and 9 : “The error bars correspond to statistical (standard deviation) errors”. In the caption of Fig.5 we added “statistical (standard deviation)” before “error bars”. In the caption of Fig. 7, we added “The vertical error bars correspond to statistical (standard deviation) errors”.

Reviewer 3 (former comment): In Figure 5 are shown the results for $m/z = 115$ (top), 70 (middle) and total (bottom row). Why are in the total data so many points missing? Here are basically the data from $m/z=70$ shown. And even the PES is an exact copy of the middle row (at least for left and middle column).

Former Authors answer: The b_1 curves correspond to a normalized photoelectron angular distribution, ie to a normalization to the total cross section, obtained by dividing the raw dichroism data by the total (PES) intensity. Therefore, as the PES intensity vanishes, the b_1 values acquire huge error bars and are essentially meaningless (-> division by zero). This is why, for the sake of clarity, we have used an algorithm that suppresses the b_1 values plotting in the baseline regions outside the principal peak areas, with a 10 % threshold in general (15 % in some cases where the S/N is lower) with respect to the maximum PES intensity. [The value and need for such a procedure is already hinted on Fig.4c, for instance, when error bars clearly start to quickly increase to lower binding energies as the PES intensity starts to vanish.] We added in the caption of Fig. 5 : “PECD points corresponding to a relative PES intensity below 10 % (15% for the m/z 115 TD₄₁₅ plot) of the maximum signal are not shown,

due to their increasing error bars when normalized by the PES intensity”.

The reason for which we chose to apply a 15 % (instead of 10 %) cut-off threshold for the m/z 115 TD_{415} plot of panel (a) is obvious when looking at the figure below in which the plot has been done with a 10 % threshold. This clearly leads to the presence of 3 non-significant points affected by a huge error bar around 9.3 eV ionization energy. These points are meaningless and does bring any information since they correspond to vanishing cross sections (PES). To be consistent, we also added in the caption of Fig. 4: “PECD points corresponding to a relative PES intensity below 10 % of the maximum signal are not shown, due to their increasing error bars when normalized by the PES intensity.” The fact that the $m/z=70$ and the total data (PES and b_1 curve) are very similar is normal since by far the m/z 70 fragment dominates the mass spectrum at 10.2 eV, the parent (m/z 115) representing a very minor component of the mass spectrum (see Fig. S4), ie of the “total” signal.

Reviewer 3 (new comment): I agree in general. However the authors could also combine 3 data points to one (as they would have done anyways, if the photon energy resolution would be not as good as they had). Therefore the intensity per point is about 3 times larger, the error bars $\sqrt{3}$ smaller and the data could be presented. Obviously leaving out points gives the readers a bitter aftertaste.

New Authors answer: The reviewer is confusing here (again) binding energies and photon energies. Here what matters is only the resolution in terms of kinetic energy of the electron (by far dominating). Besides, the binning procedure the reviewer is suggesting be applied will never compensate the huge error bar linked to a division of the signal by essentially a vanishing PES value. So we prefer leaving the figure as it is, in a form which is a standard practice in PECD literature.

Reviewer 3(former comment): while a fit to experimental data is shown. How can results be fitted, without data points?

Former Authors answer: In the text p14 we describe the fitting: “To pursue this more quantitatively, we have taken an expression, $b^{\text{expt}} = x \times b^{\text{C}} + 1 - x \times b^{\text{D}}$, applying a least squares fit for x to each data set”. We are thus performing a 1-parameter linear least squares fit of a model function (weighted combination of two pre-calculated curves) to a minimum of 5 experimental points for each sample condition. It may help to recognise that this is fully analogous to e.g. a least squares straight line fit to discrete data points, where the modelled best fit line can be used to both interpolate between experimental points, and to extrapolate beyond them. Of course, in any such case one’s confidence depends not just on the goodness of fit, but on the validity of the presumed model function.

Reviewer 3 (new comment): In general fine with me. However the color coding and description of the broken lines suggests that the lines have more in common with the data points than they actually have. The broken lines are basically an average of the theory curves folded with a value extracted from the data points. The lines are neither pure theory nor pure experiment. In order to avoid misinterpretation here and being

fair presenting results, I ask the authors putting the explanation of the broken lines separated from the points in the inset box.

New Authors answer: The figure caption is absolutely clear, unambiguous and fair about this:

“The mean experimental $b_1^{\{+1\}}$ data points, given by symbols, Also included are solid curves showing the CMS-X α calculations for conformers C and D. Best fits to the experimental data sets obtained by combining these predictions (see text) appear as broken lines”.

The purpose of a legend in the figure itself is to provide a key identifying plotted items according to symbol/line/colour in a concise manner – *not* to provide an “explanation” of the items. This it does, while the caption specifies the distinction between experimental values, calculations, and fits. The body of the text of course offers the full description and explanation.

Reviewer 3 (former comment): Similarly in Figure 8. Where are the data points for green/blue at higher kinetic energies? They have obviously been recorded, as Figure 2 suggests. And in Figure 2 it would be nice to put some numbers on the y-axis - just to better show that the data are shown on a linear scale.

Former Authors answer: The answer is again that no measurements were made above 13.5 eV photon energy with the TD (blue and green data), ie above 4 eV electron KE for Conformer Type II (allowing for its 9.65 eV binding energy). Figure 2 shows TPEPICO data which are obtained by scanning the photon energy and recording only threshold electrons (Zero KE.). Contrastingly, PECD data from Fig. 7 & 8 are obtained by recording all electron energies but only at a few fixed photon energies. Figures 2 and 8 thus show the results of very different measurement techniques and the suggested direct comparison is invalid. In particular the comment appears to be confusing the “Photon Energy” axis in Fig.2, and the “Electron Energy” axis in Fig.8. The ~4 eV upper limit (electron energy) of the blue/green data sets in Fig. 8 in fact corresponds to a photon energy of 13.5 eV once the 9.65 eV ionization energy is allowed for. As for the linear scale of Fig.2, there the caption clearly notes “The relative signals have been normalized according to the maximum of the TPES first band centred around 9.5 eV” (for the purposes of facilitating comparison). Thus, the absolute intensity scale for the 3 spectra in each subplot is different. The axes are consequently presented in arbitrary units, but linearly spaced ticks indicating a linear scale. This is very common practice.

Reviewer 3 (new comment): I agree with the authors that, as done in Fig. 2, it is common practice to present in arbitrary units. However, it is also common practice to have written a number on this axis (as “arbitrary units” doesn’t mean anything); just put 1, 2, 3, 4 on the y-axis and its fine.

New Authors answer: In Photoelectron spectroscopy, the common practice is not to

put any vertical axis labels since they are meaningless (just relative not normalized intensities). Adding any labels would be misleading.

Also I do not understand than why in Figure 7 the 3rd data point is “missing” (where blue/green are shown), while in figure 8 the 2nd data point is missing? For this, please see figure below.

New Authors answer: As we discussed in the previous rebuttal letter, the point at 10.8 eV photon energy for the RH conditions has not been measured, therefore cannot be plotted. This corresponds to the 3rd point in Fig. 7 (~2.1 eV KE for conformer type I) and to the second point of Fig. 8 (~1.2 eV KE for conformer type II).

Another point. The distinction between the conformers type I and II is done by the electron’s kinetic energy. This is basically an offset of 0.5 – 0.7 eV as can be seen from table 1. This is also roughly the difference between the right most point in fig. 7 and 8. But why is this not a constant offset, rather than an increasing offset, when comparing 7 and 8 (see figure below)?

New Authors answer: Here the reviewer is making a mistake in comparing the nth point of Fig. 7 with the nth point of Fig.8. Indeed for conformer type II there is no measurement at 9.5 eV (corresponding to the threshold of conf II), this is why the “missing point” mentioned above corresponds to the 3rd point of Fig 7 and 2nd point in Fig. 8. If one compares the nth point of Fig 7 (Type I) with the (n-1)th of Fig.8 (conf II) then there is, of course, as expected, a constant kinetic energy offset between the two conformer types (of 0.9 eV).

Reviewer 3 (former comment): It is essential to show that the molecule, which is investigated is safely brought intact into gas phase and not destroyed.

Former Authors answer: We believe that the characterization of gas phase proline in the various vaporization conditions is already well discussed and documented in the Methods section and associated bibliography. Plus, we already address the thermal decomposition issue in response to reviewers 2 (point 2) in the former rebuttal letter: “Regarding Fig S4 (former S1), an increase of temperature favors the fragmentation of the cation in a hot ground state (a very well documented effect of dissociative photoionization, see Ref. 71 and 72), not a thermal decomposition of the neutral (which we never observed with the TD method for any amino-acids). Would the latter process occur, the m/z 70 peak in the RH panels (c) would have a double structure: a sharp peak (for neutral decomposition) superimposed onto a broad one corresponding to the genuine dissociative ionization of intact neutral proline (KER effect). This is not the case. Also, by filtering the electron image with various part of this peak (center or wings) we find the same associated PECD, meaning that we are sampling the same neutral molecule: the intact parent Pro”

Reviewer 3 (new comment): OK, I missed this point from reviewer 2. And I agree

that for RH the molecule stays at least intact, which is obviously not the case for TD, where the peak is much broader and additionally shows higher masses (from clusters/dimers, which asymmetrically dissociated. From H₂O dimers/clusters it is well known that in a mass spectrum one dominantly observes HO⁺ and (H₂O)_nH⁺). The authors haven't ruled out problems with clusters.

New Authors answer: We already answered this point above in great detail in our previous rebuttal letters: there is no thermal decomposition by using the TD method (this is why we use it !), and there cannot be any clusters formed by aerosol thermodesorption since the density and internal temperature of the neutral parent (384-452 K) are respectively too low and too high to form any neutral clusters. Again, the shoulder on the long-TOF side of peaks in the mass spectra of fig 11 are due to electrostatic effects linked to the size of the thermodesorption plume (see below), and cannot be associated to protonated clusters fragments which would in addition be centred at mass m+1 and not at ~ m+4 as in fig. 11.

Reviewer 3 (former comment): The mass specs in Figure 11 (a) and (b) clearly indicate that the hump towards higher masses that dimers/clusters are created, which then fragment -> Thus a larger mass, depending on the additional group is observed. And even the correct mass of a molecule, doesn't prove that the molecule is present in its shape/conformer, as expected. I miss a clear discussion on this point.

Former Authors answer: The mass spectra of Fig.11 (a) and (b) corresponds to TD conditions. As mentioned in the caption of Fig.11, the broad and asymmetric shape of the peaks (both parent and fragments) are simply due to the very large interaction zone when using the TD associated with the inhomogeneous electric field used in the photoionization source for the electron VMI detection. This is not at all associated to the presence of near-monomer-mass fragments formed from unstable dimers/cluster. Actually, the energy releases that would be necessary to account for the observed peak broadening in such a scenario would be completely unfeasible (several eV). We performed checks that these broadened peaks do not in fact encompass varying masses/structures by comparing the electron spectra (PES and PECD) recorded in coincidence with ions sampled at different points across the peak widths. The origin of the ToF peak broadening seen with the TD sample condition have been clarified by the addition of two sentences to the "Characterisation of vapour" section of the manuscript (p.23): "The TD spectra show some asymmetric instrumental peak broadening due to ions from the spatially extended plume of desorbed sample experiencing a greater variability in the non-uniform source extraction field (required for the VMI electron detection). Checks were performed to establish that these broadened peaks are in fact homogeneous (do not encompass varying masses/structures) by examining the electron images (and hence PES and PECD spectra) recorded in coincidence with ions sampled at different points across the peak widths." And then adding to the next sentence that the RH ToF spectrum "..., which is not so afflicted thanks to a tightly collimated molecular beam source." Regarding the possibility of generating clusters with the TD method, we already answered this point in the first rebuttal letter (point 4 of reviewer 2), the answer is pasted below: " It is true that the presence of clusters would affect the PES (and the PECD profile in fact). This is a well-documented effect (see ref 28 (now Ref. 34) for a collection of references on clustering effect on PECD). The

usual effect on the PES is a red- shifting on the parent mass (due to cascading dissociation ionization processes) which we did not observe, nor in any trace of stable dimer at m/z 230. This is not surprising at all since even with the TD493 method (providing a plume of neutral not a molecular beam per se) the internal temperature was determined at 384 K (see Table 2), a temperature at which no Van der Waals nor H-bonded clusters can be formed and survive (this would require a quite strong adiabatic expansion leading to much colder temperatures to be formed, see for instance Nahon et al. Phys. Rev. A 82, 032514 (2010) or Powis et al., PCCP 16, 467 (2014)).”

REBUTTAL

Reviewer 3 (new comment): I don't know the setup exactly and therefore cannot check that “large interaction zone when using the TD associated with the inhomogeneous electric field used” is the source. A check with Argon/Krypton would help here. However, it looks strange that the peak is asymmetric with a shape that rather looks like two peaks, with one being factor 2 larger than the second one, which is shifted by a few amu. The water peak in Figure S4 TD493 at 13.5 eV has much less of a low plateau on the right side. And this little plateau is also present in the RH data, which suggests that not the TD-technique is solely the source of the broadening.

New Authors answer: Well... unlike the reviewer, we know very well our set-up and the corresponding electrostatic configuration and we extensively answered this point in the previous rebuttal letter: the origin of the shoulder is electrostatic and not due to some clusters or other species since as we stated in page 23 : “Checks were performed to establish that these broadened peaks are in fact homogeneous (do not encompass varying masses/structures) by examining the electron images (and hence PES and PECD spectra) recorded in coincidence with ions sampled at different points across the peak widths”. Using argon or krypton instead of helium as a nebulizing gas would not change anything (this is just a nebulizing gas to form aerosol from a solution, not a carrier gas as in a molecular beam). Neither would be recording mass spectra of rare gases because they would not come directly from the plume. They would actually show the same shapes as the water or N₂ peaks observed in Figure S4. These shapes have always been found in previous experiments with TD.

As for the water peak in Fig. S4 TD 493, showing indeed a much more limited plateau on the right-hand side, the explanation is very simple. The water comes mainly from the gas phase thermal water present in the chamber as a background gas, not from aerosols desorption (unlike proline) so it is differently affected by the electrostatic conditions. The very same explanation holds for the RH case.

Also I agree that the broadening of the peak does not originate from the kinetic energy of a dissociating (parent/fragment) ion. There is not enough energy, as the authors also said, and it would make the peak symmetrically broader. In Tia et al. J. Phys. Chem. Lett. 4, 2698-2704 (2013). Fig. 1b there is for alanine with the TD technique also a broader peak shown, but there the amount of “larger” masses is much less than in the present paper, or in J. Phys. Chem. Lett. 12, 2385-2393 (2021) Fig. 1 where Serine molecules are TD and no asymmetric peak broadening at all is visible. This suggests that there more than purely a problem of the imaging.

New Authors answer: The shape of the TOF peaks in the mass spectra strongly depends on the electrostatic conditions and especially on the distance between the tip of the TD and the synchrotron axis.

For the alanine case (Tia et al 2013) the TD was placed a bit further away from the SR axis, so that the shoulder is indeed slightly reduced. The case of our recently-published paper on serine (Hartweg et al. J. Phys. Chem. Lett. 12, 2385 2021) is different, since there we found a setting of the TD that allowed a further ROI filtering in the ion image (as we could achieve here from the RH conditions up to 11.5 eV but not for the TD cases), leading to a clear sharpening of the peak TOF.

But again, this issue of the peak shape is a marginal issue as long as we checked the homogeneous character (see comment page 23) peaks and as long as we are not aiming at separating closely lying mass peaks, which is our case here for proline (we want to separate m/z 115 from m/z 70 basically).